# Pumilio protects Xbp1 mRNA from regulated Ire1-dependent decay

Fátima Cairrão [1✉], Cristiana C. Santos[1], Adrien Le Thomas[2], Scot Marsters[2], Avi Ashkenazi[2] & Pedro M. Domingos [1✉]

The unfolded protein response (UPR) maintains homeostasis of the endoplasmic reticulum (ER). Residing in the ER membrane, the UPR mediator Ire1 deploys its cytoplasmic kinase-endoribonuclease domain to activate the key UPR transcription factor Xbp1 through non-conventional splicing of Xbp1 mRNA. Ire1 also degrades diverse ER-targeted mRNAs through regulated Ire1-dependent decay (RIDD), but how it spares Xbp1 mRNA from this decay is unknown. Here, we identify binding sites for the RNA-binding protein Pumilio in the 3′UTR *Drosophila Xbp1*. In the developing *Drosophila* eye, Pumilio binds both the Xbp1[unspliced] and Xbp1[spliced] mRNAs, but only Xbp1[spliced] is stabilized by Pumilio. Furthermore, Pumilio displays Ire1 kinase-dependent phosphorylation during ER stress, which is required for its stabilization of Xbp1[spliced]. hIRE1 can phosphorylate Pumilio directly, and phosphorylated Pumilio protects Xbp1[spliced] mRNA against RIDD. Thus, Ire1-mediated phosphorylation enables Pumilio to shield Xbp1[spliced] from RIDD. These results uncover an unexpected regulatory link between an RNA-binding protein and the UPR.

[1] Instituto de Tecnologia Química e Biológica, Universidade Nova de Lisboa, Av. da República, 2780-157 Oeiras, Portugal. [2] Cancer Immunology, Genentech, Inc., 1 DNA Way, South San Francisco, CA 94080, USA. ✉email: cairrao@itqb.unl.pt; domingp@itqb.unl.pt

Metazoan cells respond to endoplasmic reticulum (ER) stress by activating an intracellular network of signaling pathways, known as the unfolded protein response (UPR)[1,2]. In higher eukaryotes, the UPR involves three ER transmembrane transducers: inositol-requiring enzyme 1 (IRE1), pancreatic ER kinase (PKR)-like ER kinase (PERK), and activating transcription factor 6 (ATF6). When misfolded proteins accumulate in the ER, IRE1 activates the downstream transcription factor X-box binding protein 1 (XBP1), via the non-conventional splicing of Xbp1 mRNA[3–7]. The cytoplasmic kinase- endoribonuclease domain of IRE1 mediates the splicing of a 26-nucleotides long intron from the Xbp1 mRNA, causing a frame-shift during translation that introduces a new carboxyl domain in the XBP1 protein. The resulting spliced form of Xbp1, XBP1[spliced], is a functionally active transcription factor that upregulates the expression of ER chaperones and other UPR target genes[8].

The intracellular localization, stability and translation of mRNAs is regulated by interaction of RNA-binding proteins (RBPs) or microRNAs with specific sequences present in the 3′ untranslated regions (3′UTR) of the mRNA[9–11]. In budding yeast, the 3′UTR of the Xbp1-orthologue Hac1 targets Hac1 mRNA to foci of activated IRE1 at the ER membrane, which enables IRE1-mediated splicing of Hac1 mRNA[12]. In contrast, in mammalian cells the 3′UTR of Xbp1 seems to be dispensable for the targeting of Xbp1 mRNA to activated IRE1[13]. Instead, the targeting occurs through the tethering of 2 hydrophobic regions (HR1 and HR2) present in the XBP1[unspliced] protein to the cytosolic side of the ER membrane[13,14]. This latter process involves a tripartite complex comprising Xbp1 mRNA, a ribosome, and the nascent XBP1 protein, and additionally requires translational pausing of the Xbp1[unspliced] mRNA[14].

Besides Xbp1 mRNA splicing, the endoribonuclease (RNase) domain of Ire1 also performs a function known as regulated Ire1-dependent decay (RIDD), which mediates the depletion of specific mRNAs[15] and/or microRNAs[16]. While in Drosophila cells RIDD degrades ER-targeted mRNA relatively promiscuously, in mammalian cells it is thought to depend on a specific Xbp1-like mRNA sequence endomotif within a stem-loop structure, and on translational state of the mRNA target[17]. The phosphorylation and oligomerization state of IRE1 plays an additional role in controlling IRE1's RNase activity[18]. RIDD can act as a post-transcriptional mechanism to deplete mRNAs, thereby affecting both ER homeostasis and cell fate[19–22].

RNA-binding proteins (RBPs) constitute an important class of post-transcriptional regulators. RBPs are involved in multiple critical biological processes, relevant to cancer initiation, progression, and drug resistance[23]. Several recent studies validate the role of the Pumilio family of RBPs in diverse biological processes in several organisms. Specific mRNA targets of two human Pumilio isoforms (PUM1 and PUM2) include oncogenes, tumor suppressors, and other factors implicated in oncogenic and cell death pathways[24–26]. In several organisms, PUM proteins also play a conserved role in stem cell proliferation and self-renewal[27–30]. PUM proteins are essential for development and growth, and their dysfunction has been associated with neurological diseases, infertility, movement disorders, and cancer[31–36].

Drosophila Pumilio–the founding member of the PUM family—is characterized as a translational repressor and is involved in embryo patterning, fertility and the regulation of neuronal homeostasis[31,37–40]. PUM proteins act as post-transcriptional regulators, by interacting with consensus sequences called Pumilio regulatory elements (PRE) in the 3′UTRs of target mRNAs[9] to modulate their translation and/or degradation[41,42]. Recent findings support a direct role of PUM proteins in the activation/protection of specific RNAs[26]. PUM proteins are composed of distinct functional domains: N-terminal repressor domains (N), which are unique to different PUM orthologues, and a Pumilio homology domain (PumHD), which recognizes the PRE. These domains mediate the normal repressive role of PUM proteins by antagonizing the translational activity of Poly(A) binding protein (PABP)[42,43]. The PUM N terminus is required to fully rescue developmental defects of a pum mutant[44,45].

Although much progress has been made toward understanding the biological roles of PUM proteins, it remains to be determined how they regulate their target mRNAs particularly under diverse biological conditions such as cellular stress. Here, we uncover an unexpected functional link between Pumilio and the Ire1-Xbp1 pathway. We show that Pumilio protects Drosophila Xbp1 against RIDD, without perturbing canonical Ire1-driven Xbp1 splicing. Furthermore, we provide evidence that Ire1 phosphorylates Pumilio under ER stress and that this is essential for Pumilio-mediated protection of Xbp1 mRNA. These results identify an important regulatory mechanism connecting RBPs and the UPR.

## Results

**The 3′UTR of Xbp1 contains cis elements that regulate mRNA stability.** The 3′UTRs of Drosophila Xbp1[unspliced] and Xbp1[spliced] differ in the length, the former being 600 bp longer than the latter (Fig. 1a). We conducted a search on the 3′UTR of Drosophila Xbp1 mRNA and identified two putative Pumilio regulatory elements (UGUAXAUA) (Fig. 1a, b). To test experimentally if these putative PREs can regulate mRNA, we constructed reporters containing GFP, expressed under the control of the metalotheionin promoter[46,47], together with the 3′UTR of Xbp1 in wild type or a mutated version of each of the two PREs (Fig. 1b). To examine the impact of these PREs on mRNA stability, we mutated the sequence encoding the PRE (UGUAXAUA) to the non-functional element ACAAXAUA (Fig. 1b).

The mRNA decay of the resulting reporters was analyzed in transiently transfected Drosophila S2 cells, which were treated with actinomycin D to stop transcription. For each mutated version of the PREs the stability of the GFP reporter decreased as compared to the wild-type 3′UTR, suggesting a role for these regulatory regions in the post- transcriptional regulation of Xbp1. We also compared the effect of the 3′UTRs of Xbp1[spliced] and Xbp1[unspliced] forms by quantitative RT-PCR, which showed a greater effect of the former 3′UTR on stability of the GFP reporter mRNA (Supplementary Fig. 1a). This result suggests that other, yet unidentified regulatory regions of RNA stability may be present in the 3′UTR of Xbp1[unspliced].

**Pumilio binds Xbp1 mRNA in vitro and in the developing Drosophila eye.** To confirm the interaction between Pumilio and Xbp1 mRNA, we tested the binding affinity of Pumilio to one of the PRE elements present in the Xbp1 3′UTR by electrophoretic mobility shift assays (EMSA - Fig. 1c). For this experiment we used purified protein of human PUM1, containing the Pum-HD domain (Supplementary Fig. 2a), which shares 80% of sequence conservation to Drosophila Pum-HD. Interaction between hPUM1-HD and the PRE1 in the Xbp1 3′UTR was tested using a range of protein concentration (100-300 nM) and a biotinylated RNA oligo containing the WT PRE1 sequence (5′-N5-UGUA-CAUA-N12-3′BioTEG). hPUM1-HD binds the RNA oligo, as seen by appearance of a slower migrating band in the native gel, corresponding to the RNA-protein complexes. The binding selectivity of hPUM1-HD to the RNA oligo was examined by the presence of increasing concentrations of an unlabeled RNA oligo competitor, containing the WT or the mutated (UGUA to ACAA) target sequence for Pumilio binding. Although the shifted

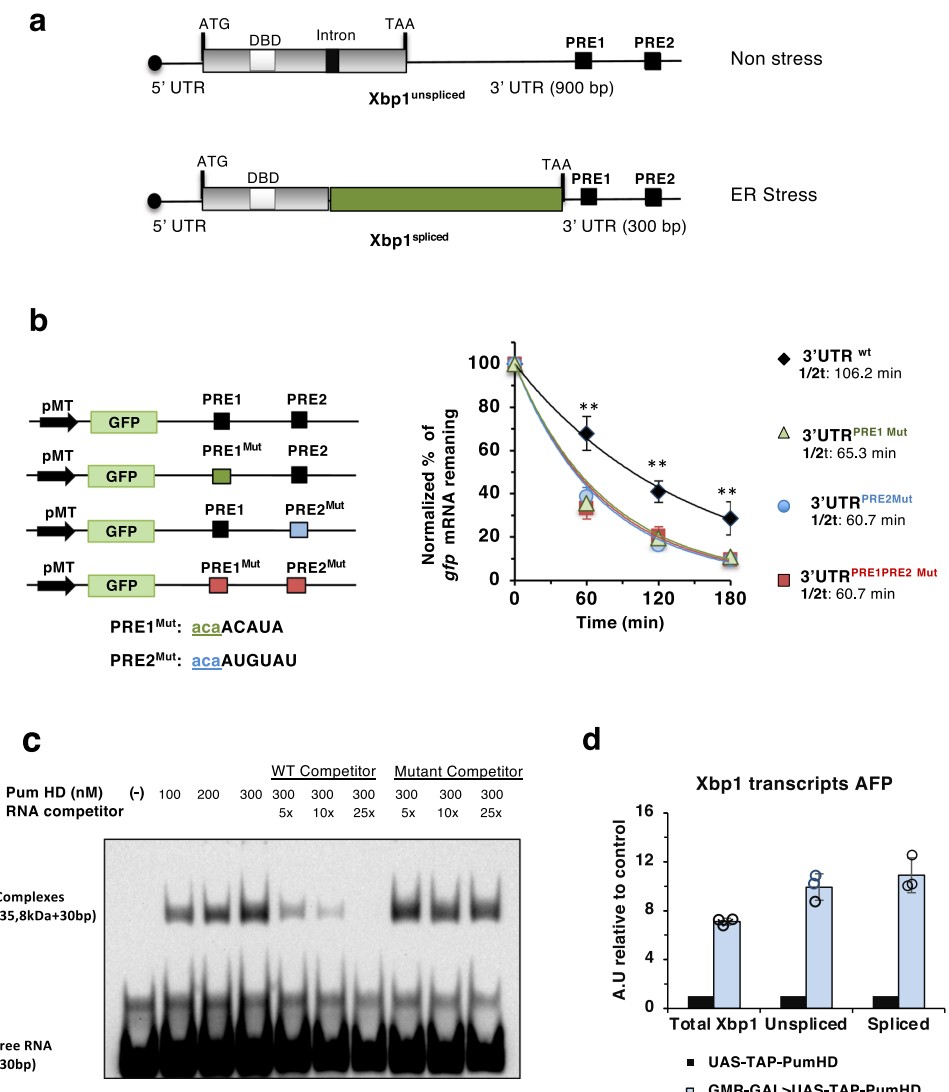

**Fig. 1 The 3′UTR of Xbp1 contains cis elements of mRNA stability. a** Schematic representation of *Drosophila* Xbp1 transcripts (unspliced and spliced). ATG (start codon), TAA (stop codon), DBD (DNA binding domain), Intron (23-nt hairpin structure) recognized by Ire1 RNase domain, bp (base pairs). Upon ER stress, Ire1 removal of the intron causes a frame shift to generate Xbp1spliced. Two Pumilio regulatory elements (PRE1 and PRE2) are present in the 3′ UTR of Xbp1. **b** Xbp1 3′UTR pRM GFP reporters bearing the WT and mutant PREs from the Xbp1 3′UTR. The consensus binding sites UGUA for Pumilio were mutagenized into ACAA for PRE1Mut and PRE2Mut; pMT, metallothionein promoter. Stability of the GFP reporters was assessed after actinomycin D addition [5μg/ml], by quantitative RT-PCR (qRT-PCR), using primers specific for *gfp* and *rp49* mRNAs (control). Levels of mRNA reporter were normalized to those of *rp49* mRNA. Data are presented as mean ± SD. $n = 3$ of biologically independent experiments. wt, black; PRE1Mut, green; PRE2Mut, blue; PRE1PRE2Mut, red. One-way ANOVA followed by Tukey´s HSD multiple comparisons test was used to calculate significant differences between wt and mutant PREs. 60 min: $**p = 0.01$; 120 min: $*p = 0.03$, $*p = 0,015$; $**p = 0.009$; 180 min: $*p = 0.029$, $*p = 0,026$, $*p = 0,018$. **c** Electrophoretic mobility shift assay (EMSA) showing binding of purified hPUM1-HD, with the PRE1 of the Xbp1 3′UTR. hPUM1-HD (38,5 kDa) binds the RNA oligo containing the WT PRE1 sequence (30 nt), as seen by appearance of a slower migrating band corresponding to the RNA-protein complexes. The binding selectivity of hPUM1-HD to the RNA oligo was examined by the presence of increasing concentrations of an unlabeled RNA oligo competitor (30nt), containing the WT or the mutated (UGUA to ACAA) target sequence for Pumilio binding. $n = 2$ of independent experiments. **d** Quantitative RT-PCR using primers specific for Xbp1spliced, Xbp1unspliced, and total Xbp1 transcripts, from GMR-GAL4>UAS-TAP-PumHD RNA pull downs, normalized against levels in the UAS-TAP-PumHD control pull downs. Results show interaction between Xbp1spliced and Xbp1unspliced transcripts and PumHD. Data are presented as mean ± SD. $n = 3$ replicates from one RNA pulldown assay. Source data for figures b-d are provided as a Source Data file.

band corresponding to the RNA-PUM complex disappeared following the addition of the WT competitor oligo, the addition of the mutant competitor oligo yielded no effect on the binding of hPUM1-HD to the labeled WT Xbp1 oligo.

To analyze if Pumilio could act as a regulator of Xbp1 in fly tissues, we expressed a transgene encoding a TAP-tagged[48] Pumilio RNA-binding motif (UAS-TAP-PumHD), under control of the eye-specific GMR-GAL4 driver. We conducted TAP pull-down RNA affinity purifications from heads of transgenic adult flies with the genotype GMR– GAL4>UAS-TAP-PumHD, and performed RT-PCR to determine whether the endogenous Xbp1 mRNA was enriched in flies over-expressing TAP-Pum-HD (Fig. 1d, Supplementary Fig. 1b). We found that PumHD could bind equally well to the 3′UTR of Xbp1unspliced and Xbp1spliced (Fig. 1d). These results suggest that endogenous Xbp1 transcripts in the *Drosophila* eye can be regulated by Pumilio.

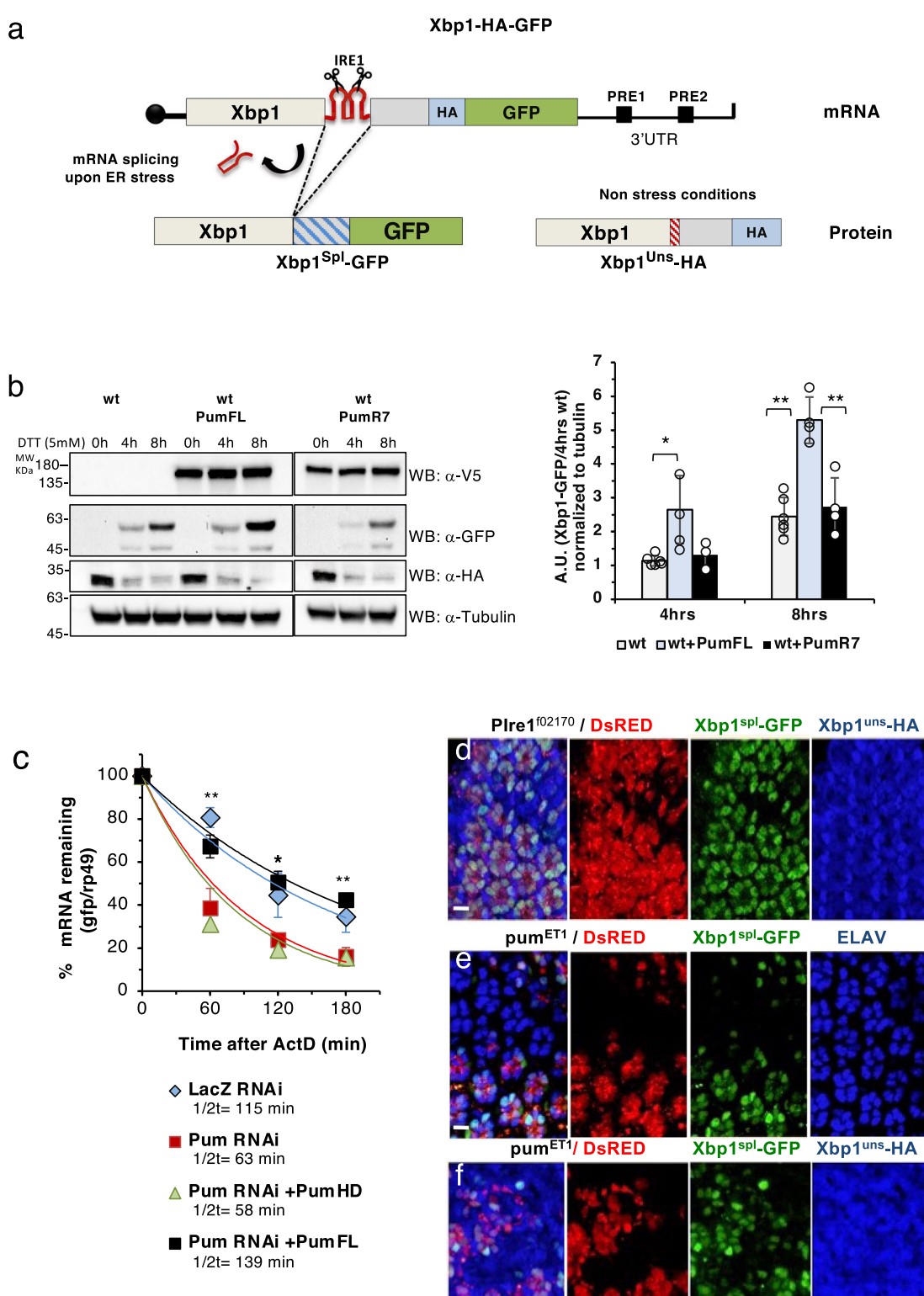

**Pumilio regulates the stability of Xbp1^{spliced} mRNA**. The activity of Ire1 in mediating Xbp1 splicing can be assayed with an Xbp1-GFP reporter[49], wherein Xbp1^{spliced} is tagged with GFP. We constructed a new Xbp1 reporter (Xbp1-HA-GFP) (Fig. 2a), which distinguishes the translation product of Xbp1^{unspliced}, fused to an HA tag, from that of Xbp1^{spliced}, fused to GFP. We first tested the expression of the reporter translation products by

western blot under non stress and stress conditions in the presence of dithiothreitol (DTT), an ER stress inducing chemical (Fig. 2b). In this context, overexpression of full-length Pumilio[50] led to an increase in Xbp1^{spliced}, but Xbp1^{unspliced} protein levels remained unchanged in comparison with controls (transfected cells with Xbp1-HA-GFP only). Overexpression of PumR7[50], a Pumilio mutant defective for RNA binding, did not cause a

**Fig. 2 Pumilio regulates the stability of Xbp1$^{spliced}$ mRNA. a** Schematic representation of the Xbp1-HA-GFP reporter construct. This reporter allows the detection of Xbp1$^{Spl}$ (spliced) and Xbp1$^{Uns}$ (unspliced) forms. pUAST-Xbp1-HA-GFP contains a HA-tag in frame with Xbp1$^{unspliced}$. Upon ER stress, the Ire1 mediated splicing causes a frame shift so that GFP becomes in frame with Xbp1$^{spliced}$. **b** Expression of Xbp1-HA-GFP reporter in S2 cells in the absence (0h) and presence of ER stress (4h and 8h DTT treatment). Xbp1$^{spl}$-GFP (56kDa) is detected under stress conditions. Overexpression of Pumilio (Pum-V5) but not PumR7 (Pum mutant defective for RNA binding) increases the levels of Xbp1$^{spl}$-GFP, without affecting Xbp1$^{uns}$-HA (29,8 kDa). WT(white bars): S2 cells transfected with Xbp1-HA-GFP plasmid; PumFL(blue bars): Xbp1-HA-GFP + plZ-Pum-Full length-V5; PumR7(black bars): Xbp1-HA-GFP + plZ -PumR7-V5. Quantification for Xbp1$^{spl}$-GFP is presented as mean ± SD, normalized to tubulin levels and to the levels of WT(4h). One-way ANOVA coupled with Tukey's post hoc tests: *$p = 0.02$ at 4 h for wt($n = 6$) vs wt+PumFL($n = 4$); non-significant between wt+pumR7 ($n = 3$) and wt or wt + PumFL. **$p = 0.001$ at 8 h for wt($n = 6$) vs wt+PumFL($n = 4$) and **$p = 0.002$ for wt+PumFL vs WT+ PumR7($n = 4$). n represents the number of biologically independent experiments. **c** Stability of Xbp1-HA-GFP was compared between control S2 cells (LacZ RNAi) and cells treated with Pum RNAi. Inactivation of Pumilio destabilizes Xbp1-HA-GFP (red squares). Full length Pumilio (Full Pum, black squares) restores the stability of Xbp1-HA-GFP, in contrast to the RNA binding domain of Pumilio (PumHD, green triangles). mRNA stability was analyzed by qRT-PCR after actinomycinD (5 μg/ml) and DTT (4 h at 5 mM) treatments. Data are presented as mean ± SD of % of mRNA remaining, normalized to rp49. $n = 3$ of biologically independent experiments. One-way ANOVA coupled with Tukey's post hoc tests for each time point. At 60′ (LacZ RNAi and PumFL vs PumRNAi,PumHD,**$p = 0.0010$); at 120′ (PumFl vs PumRNAi **$p = 0.0065$, vs PumHD **$p = 0.0045173$); at 180′ (PumFl vs PumRNAi), **$p = 0.0023625$ and vs PumHD, **$p = 0.0016455$. **d** The Xbp1-HA-GFP reporter is activated in the *Drosophila* photoreceptors during midpupal stages (50 h). Xbp1$^{spl}$-GFP is observed in WT photoreceptors (Red) but not in photoreceptors that are homozygous for an Ire1 null mutation (Pire1f$^{f021701}$), labeled by the absence of DsRed. Xbp1$^{uns}$-HA (blue) is observed in all cells. **e** The expression of Xbp1$^{spl}$-GFP (green) is reduced in cells homozygous for the Pum null mutation (Pum$^{ET1}$), labeled by the absence of DsRed expression, in comparison with wildtype cells (DsRed positive). Elav (blue) was used as a marker of the photoreceptors. **f** The expression Xbp1$^{spl}$-GFP (green) is reduced but Xbp1$^{uns}$-HA (blue) is unaltered in cells homozygous for Pum$^{ET1}$, labeled by the absence of DsRed expression, in comparison with wildtype cells (DsRed positive), indicating that the regulatory role of Pumilio is only on the Xbp1$^{spl}$ form. Anti-HA was used to label Xbp1$^{uns}$-HA protein. Scale bars represent 10 μm. $n \geq 10$ mutant eyes for **d**–**f**. Source data for figures **b**–**f** are provided as a Source Data file.

change in Xbp1$^{spliced}$ or Xbp1$^{unspliced}$ levels, in comparison with controls.

Subsequently, we determined the stability of the Xbp1-HA-GFP reporter mRNA in transiently transfected S2 cells, either with Pumilio RNAi or control LacZ RNAi depletion (Fig. 2c, Supplementary Fig. 1c). Depletion of Pumilio decreased the stability of the Xbp1-HA-GFP reporter transcript as compared with controls (Fig. 2c). When Pumilio levels were restored by co-transfecting S2 cells depleted of Pumilio with a plasmid expressing either the full-length Pumilio protein (Pum-FL)[50] or the Pumilio RNA binding domain (Pum-HD), the mRNA stability of Xbp1-HA-GFP reporter returned to the levels of the control experiment (LacZ RNAi treatment), but it did so only when the full-length protein was present (Fig. 2c). These results indicate that Pumilio has a protective role against degradation of the Xbp1 mRNA, and that although the HD region confers specific RNA interaction with the Xbp1 transcripts, the full-length Pumilio protein is required for its protective role.

**Pumilio regulates Xbp1$^{spliced}$ during photoreceptor differentiation**. Next we tested the regulation of Xbp1 mRNA by Pumilio in a physiological, developmental context. The Ire1 signaling pathway is activated during the pupal stages in the photoreceptors[51], where it is required for photoreceptor differentiation and morphogenesis of the rhabdomere – the light-sensing organelle of photoreceptors. We generated transgenic flies with the Xbp1-HA-GFP reporter, and performed immunofluorescence analysis with antibodies against GFP and HA, to assess the expression levels of Xbp1$^{spliced}$-GFP and Xbp1$^{unspliced}$-HA in eyes containing clones of the Ire1$^{f02170}$ null mutation[51]. As expected, and validating the Xbp1-HA-GFP reporter, Xbp1$^{spliced}$-GFP expression was absent from Ire1$^{f02170}$ homozygous cells (labeled by the absence of DsRed), but Xbp1$^{unspliced}$-HA expression was unaffected in these Ire1 homozygous mutant cells (Fig. 2d). We next examined the expression of Xbp1-HA-GFP in eyes containing clones of the Pumilio null mutation pumET1[52]. In pupal eyes (50 hr), in the absence of Pumilio (labeled by the absence of DsRed), the expression of Xbp1$^{spliced}$-GFP was reduced as compared to the wild-type cells (Fig. 2e, f). However, expression of Xbp1$^{unspliced}$-HA remained unchanged (Fig. 2f),

indicating that Pumilio exerts its regulatory role on the mRNA of Xbp1$^{spliced}$-GFP but not Xbp1$^{unspliced}$-HA.

**Pumilio undergoes IRE1 kinase-dependent phosphorylation during ER stress**. It is known that Pumilio proteins may become activated to regulate target mRNAs upon phosphorylation[53]. Furthermore, the observation that the stability of the Xbp1$^{spliced}$ transcript was dependent on the full-length Pumilio protein (Fig. 2c) indicated that other domains of Pumilio, besides the RNA-binding domain, might also be important for Xbp1 mRNA stability. Recent studies[36,42,50] indicate that additional regions in the N terminus of Pumilio act as repressor domains, with two specific segments – designated Pumilio conserved motif (PCM) a and b – being found in both *Drosophila* and human PUMs (Fig. 3a and Fig. S2a). We screened the Pumilio protein sequence for potential serine/threonine phosphorylation sites using online bioinformatic tools[54] combined with phosphorylation sites reported in other screens (http://www.phosphosite.org/). We found potential phosphorylation sites across the different domains of Pumilio (Fig. 3a). To investigate if Pumilio is phosphorylated during ER stress, we conducted Phostag analysis using Pumilio proteins tagged with a C-terminal V5 epitope (Fig. 3a). It was difficult to analyze phosphorylation of full-length Pumilio[50], due to its relatively large size of ~180 kDa (data not shown). Therefore, to facilitate the analysis, we constructed 3 truncated versions of V5-tagged Pumilio (D1 = aa 1 to 547; D3 = aa 777 to 1091 and D1D3 = aa 1 to 1091), which contained the predicted phosphorylation sites.

Phostag analysis revealed that, during ER stress (4 hr DTT), Pumilio D1 displayed a 6-fold increase in phosphorylated forms, relative to non-ER stress conditions (0 hr DTT) (Fig. 3b, c). This phosphorylation pattern was confirmed by treatment with phosphatases (+CIP or + λPP) and the use of a triple phospho-mutant (M3 = T537A, S540A, S544A), which reduced the phosphorylated forms of Pumilio D1 (Fig. 3b, d).

During ER stress and after IRE1 trans-autophosphorylation, the cytosolic kinase-endoribonuclease domain of IRE1 rotates to a back-to-back configuration[2,18], possibly allowing the kinase active site to access other potential substrates. We reasoned that through binding to the 3′UTR of Xbp1, Pumilio might localize to the vicinity of the Ire1 kinase domain. Therefore, we tested whether

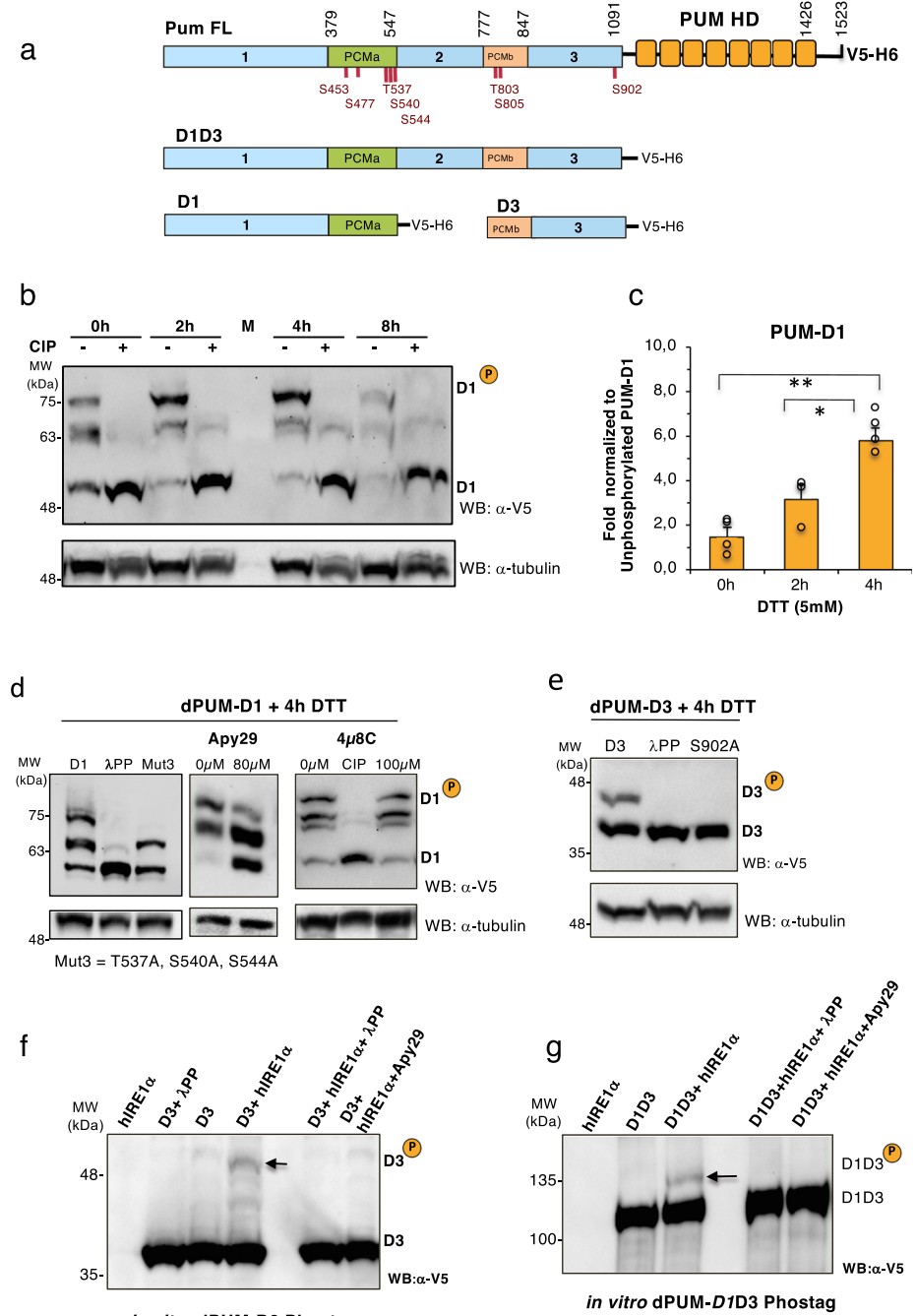

Ire1 could phosphorylate Pumilio during ER stress, using the inhibitors of Ire1 kinase activity (Apy29)[55] and compound #18[56] or an inhibitor of IRE1 RNase activity (4µ8C)[57]. Upon DTT exposure (4 h), in S2 cells transfected with PUM-D1, Apy 29 attenuated the generation of PUM-D1 hyper-phosphorylated forms (Fig. 3d). In contrast, 4µ8C did not change the pattern of PUM-D1 phosphorylation, which is expected, as 4µ8C is specific for the RNase activity and does not inhibit IRE1 kinase activity[57]. We also observed ER stress induced phosphorylation of PUM-D3, and by constructing phosphorylation site mutant versions of PUM-D3 we could identify S902 as a site that is phosphorylated under ER stress conditions (Fig. 3e).

**hIRE1 phosphorylates Pumilio and hPUM1**. To confirm that IRE1 kinase activity could directly phosphorylate Pumilio, we

performed in vitro kinase reactions using the hIRE1α kinase-endoribonuclease (KR) domain[57] and purified versions of Pumilio domains D3 and D1D3. Phostag immunoblot analysis (Fig. 3f, g) showed specific bands indicating phosphorylation of PUM-D3 and PUM-D1D3 upon incubation with hIRE1 KR. Pumilio phosphorylation was abolished upon Apy 29 or λ-PP treatment. To complement these results, we performed in vitro radioactive kinase assays using hIRE1α KR and purified versions of Pumilio and hPum1 protein domains (Fig. 4a, b and Supplementary Fig. 2b). The hIRE1α KR protein phosphorylated both Pumilio and hPum1 proteins. For negative controls, we used a kinase-dead hIRE1α KR as enzyme, or BIP (a luminal ER protein that should not be a direct target for IRE1 phosphorylation) as substrate (Supplementary Fig. 3a). Furthermore, to assure that the phosphorylation pattern observed for the PUM proteins in the

**Fig. 3 Pumilio undergoes Ire1 kinase- dependent phosphorylation during ER stress. a** Diagram of *Drosophila* Pumilio domains, with predicted phosphorylation sites indicated below in red for each amino acid residue. Truncated versions of Pumilio were constructed with V5 and 6xHis at their C-terminal end (Pum FL: Full length protein, D1D3: domain 1 to 3, D1: domain 1, D3: domain 3, PUM-HD: RNA binding Pumilio homology domain). **b** *Drosophila* S2 stable cell lines expressing PumD1 were submitted to ER stress (DTT 5 mM - 2h, 4h 8h). Phosphorylation was determined upon changes in electrophoretic mobility in SDS-PAGE using the Phostag compound (50mM). Pumilio proteins were detected by immunoblot (IB) with an anti-V5 antibody. The mobility shift of Pumilio is reversed by calf intestinal phosphatase treatment (+CIP). D1$^P$: hyper-phosphorylated forms, D1: non phosphorylated form of D1. M: protein size markers. **c** Quantification of fold increase of Pum-D1$^P$ during ER stress, normalized to non phosphorylated form (Pum-D1). Data are presented as mean ± SD of fold increase, n≥3 of biological independent experiments. One-way ANOVA coupled with Tukey's post hoc test *$p = 0.0225$ for 4 h ($n = 4$) vs 2 h($n = 3$); **$p = 0.001$ for 4 h vs 0 h ($n = 4$) from independent experiments. **d** Phostag western blot (WB) for Pumilio D1 after ER stress (DTT 5 mM, 4 h). Cells were incubated with an inhibitor of IRE1 kinase activity (Apy29, 80 µM) or an inhibitor of IRE1 RNase activity (4µ8C, 100 µM). The phosphorylation pattern of dPUM-D1 was monitored by WB with anti-V5 antibody. The phosphorylation of dPUM-D1 was reduced in a triple phosphomutant (M3 = T537A, S540A, S544A). Total protein loading was monitored by levels of tubulin. CIP (alkaline phosphatase) treatment of cell extracts after 4 h of DTT treatment. $n = 2$ of biologically independent experiments. **e** Phostag western blot (WB) for Pumilio Domain 3 (dPUM-D3) after ER stress (DTT 5mM, 4h). dPUM-D3 was detected using anti-V5 antibody. Total protein loading was monitored by levels of Tubulin. Mutation of site Ser902 to Ala (S902A) prevents the DTT induced phosphorylation of dPUM-D3. λPP: λ−phosphatase treatment of cell extracts. $n = 2$ of biologically independent experiments. **f**, **g** Phostag western blot (WB) after in vitro phosphorylation assay with purified domains of Pumilio (D3 and D1D3) incubated with human IRE1α KR (aa 464-977) in kinase buffer (2mM ATP). Controls: IRE1 alone, Pum alone (D3 or D1D3); Inhibition of phosphorylation was observed upon treatment with λ-phosphatase or the Apy29 IRE1 kinase inhibitor. Pumilio proteins were detected with mouse anti-V5 antibody. $n = 2$ of biologically independent experiments. Source data file is provided for all panels.

radioactive assays did not simply reflect IRE1 auto-phosphorylation, we used an antibody specific for phosphorylated IRE1[58]. While we observed phosphorylated monomers, dimers and oligomers of hIRE1α KR as expected, we could not detect any signal in the 40 KDa region (Supplementary Fig. 3b), which corresponds to the phosphorylated forms of Pumilio that were detected with the V5 antibody (Fig. 4b). These results show that hIRE1α KR can directly phosphorylate Pumilio.

**Pumilio protects Xbp1 mRNA from regulated IRE1-dependent decay.** We have shown that Pumilio has a protective role in the stability of Xbp1 mRNA (Fig. 2c). Ire1 can also mediate RIDD, the selective decay of ER-bound mRNAs, thereby reducing the load of nascent proteins entering the ER in order to be folded[59]. RIDD plays an import role during persistent ER stress[19] and in *Drosophila* cells it can lead to the complete degradation of even ectopic mRNAs, such as GFP[15,60]. Furthermore, Le Thomas et al.[58] identified a variety of mRNAs in human cells that do not contain the consensus stem-loop endomotif present in Xbp1 and in many canonical mammalian RIDD targets (dubbed RIDDLE – RIDD lacking endomotif), but are nonetheless bona fide targets of IRE1-dependent RNA decay. Indeed, Le Thomas *et al* showed that in its purified, fully phosphorylated oligomeric form hIRE1α KR (KR-3P) degraded not only RIDD and RIDDLE mRNA targets, but also hXBP1 mRNA. We therefore hypothesized that Pumilio may protect Xbp1 mRNA from decay by Ire1 under ER stress conditions. To test this possibility, we performed mRNA stability assays with the Xbp1-HA-GFP reporter in S2 cells treated with DTT, in the absence or presence of the Ire1 RNase inhibitor 4µ8C. The destabilization of Xbp1-HA-GFP mRNA in cells depleted of Pumilio was reverted upon 4µ8C treatment (Fig. 5a), indicating that Pumilio stabilizes Xbp1 mRNA against the RNase activity of Ire1. We also conducted mRNA stability assays using in vitro transcribed *Drosophila* Xbp1 and human XBP1 mRNAs incubated with non-phosphorylated (0P) or fully phosphorylated (3P) hIRE1α KR. Since we could not produce purified recombinant full-length Pumilio, we used different forms of hPUM1, which were pre-incubated with the Xbp1 transcript in order to assess their protection against degradation by IRE1. Full-length hPUM (hPUM1-FL) protected the *Drosophila* Xbp1 (Fig. 5b) and hXBP1 (Supplementary Fig. 4a) mRNAs from degradation by hIRE1α KR-3P, but did not block or even diminish the hairpin motif-dependent splicing of Xbp1 and hXBP1 mRNAs by KR-0P or KR-3P. By contrast, hPUM1-B (2-

827) did not protect Xbp1 mRNA from decay, consistent with the absence of the HD region necessary for binding to Xbp1 mRNA in this hPUM variant. Likewise hPUM1-C (828-1186) is incapable of such protection. Only the full-length hPUM1 protein contains all the requisite domains for protection of Xbp1 mRNA, in keeping with the above results (Fig. 2c).

**Pumilio's protection of Xbp1 mRNA is dependent on its phosphorylation state.** Having identified T537, S540, S544 and S902 as sites for Pumilio phosphorylation under ER stress (Fig. 3), we next asked if the phosphoryation status of Pumilio could regulate the effect on the stability of Xbp1$^{spliced}$. To examine this, we compared the half-life of Xbp1 upon overexpression of wt Pumilio or Pumilio phosphomutants (S902A and a quadruple mutant:T537A, S540A, S544A and S902A) in RNAi Pumilio-depleted S2 cells (Fig. 5c). As compared to wild-type Pumilio, overexpression of the phosphomutants lead to a decrease in the stability of Xbp1$^{spliced}$, demonstrating that Pumilio phosphorylation is required to promote its protective effect on Xbp1 transcripts under ER stress.

**Pumilio regulation of Xbp1 is important for the cellular response to ER stress.** Finally, we investigated the importance of this regulatory mechanism of Pumilio on Xbp1 mRNA for the response of cells under ER stress. We treated *Drosophila* larvae from the Pumilio mutation Pum$^{ET1}$ with/without tunicamycin (Tm) yeast paste food. After 20 hours of Tm treatments, around 1000 L1-L2 stage Pum$^{ET1}$ homozygous or control heterozygous Pum$^{ET1}$/TTG larva were collected and processed for RT-qPCR. The results (Fig. 5d) show that, upon tunicamycin treatment, the levels of Xbp1$^{spliced}$ and the induction of the Xbp1 targets Acat2 and Hsc3 are diminished in Pum$^{ET1}$ homozygous larva, in comparison with control heterozygous Pum$^{ET1}$/TTG larva. Similarly, we did experiments using human MDA-MB-231 cells treated with Thapsigargin (Tg) or vehicle control (DMSO). In this case the RT-qPCRs results (Supplementary Fig. 4b) reveal diminished levels for human XBP1s and the XBP1s target SYVN1, upon treatments with siRNAs for PUM1 and PUM2 (siPUM1+2).

**Discussion**
Our results uncover a novel protective effect of Pumilio on Xbp1 mRNA during ER stress. This protective effect depends on the phosphorylation status of Pumilio, which can be mediated by Ire1

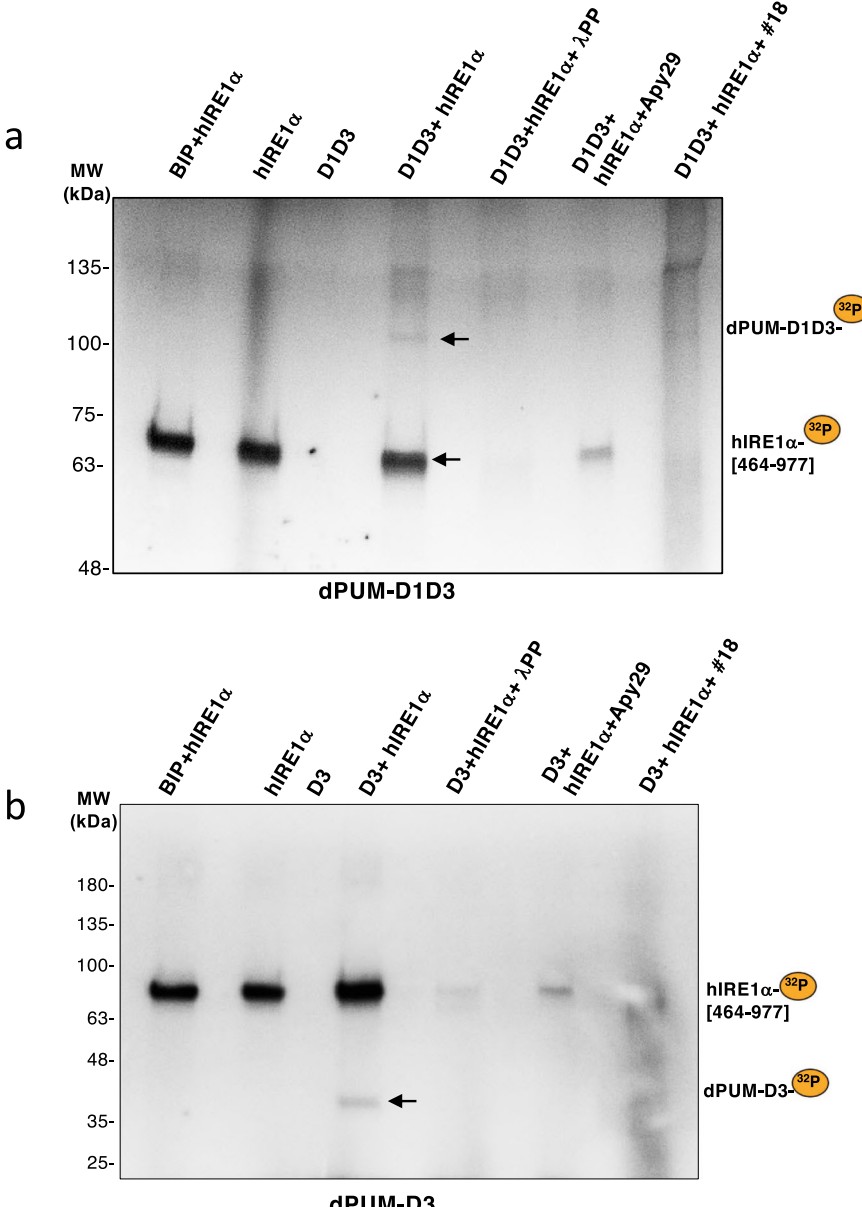

**Fig. 4 hIRE1α KR phosphorylates Pumilio in vitro.** In vitro radioactive phosphorylation assays of purified Pumilio dPUM-D1D3 (**a**) and dPUM-D3 (**b**) with hIRE1α KR (464-977). Purified proteins were incubated 2h in IRE1 kinase buffer containing γ-ATP[$^{32}$P]. Detection of radioactive dPUM-D1D3, dPUM-D3 and auto-phosphorylation of hIRE1α KR are denoted by a yellow bullet $^{32}$P. Negative controls: BIP as a luminal ER protein non-phosphorylated by IRE1, and proteins without incubation with hIRE1α KR. Specificity of phosphorylation was monitored by treatment with λ-phosphatase (λPP) or incubation with specific inhibitors of IRE1 kinase activity (Apy29 and compound #18). MW: protein molecular weights in kDa. $n = 3$ of independent experiments. Source data file is provided.

kinase activity in response to ER stress. We propose a model depicted in Fig. 6, which involves 3 steps.

In the absence of ER stress, Ire1 is not active and Xbp1 mRNA is not spliced, but Pumilio can bind the 3′UTR of Xbp1$^{unspliced}$ mRNA (Fig. 6–1). Indeed, our RNA pull down experiment (Fig. 1d) showed that Pumilio can bind equally well to the 3′ UTR of Xbp1$^{unspliced}$ and Xbp1$^{spliced}$. Accumulation of misfolded protein in the lumen of the ER causes ER stress and activates Ire1. At this stage, in order to be spliced, Xbp1 mRNA must localize in the vicinity of the cytoplasmic domain of Ire1, which harbors both kinase and endoribonuclease activities. Upon ER stress, Pumilio is phosphorylated via a mechanism that requires Ire1 kinase activity (Fig. 3). As demonstrated by our in vitro assays (Fig. 4), the kinase domain of Ire1 can

directly phosphorylate Pumilio; however, we cannot exclude at this stage the possibility that other kinases also may contribute to Pumilio phosphorylation. It is known that phosphorylation of Pum family proteins contributes to their activation and function[53]. We suggest that in this case, Ire1- dependent phosphorylation of Pumilio causes a conformational change in the mode of interaction between Pumilio and Xbp1 mRNA, such that much of the transcript, though not the canonical stem-loop structures, is inaccessible to the more promiscuous endoribonuclease activity of Ire1 (Fig. 6: step 2). This allows Xbp1 splicing to occur and furthermore permits Ire1 to carry out endomotif-directed RIDD as well as endomotif-lacking RIDDLE while sparing Xbp1$^{spliced}$ mRNA and allowing its more efficient translation (Fig. 6–3).

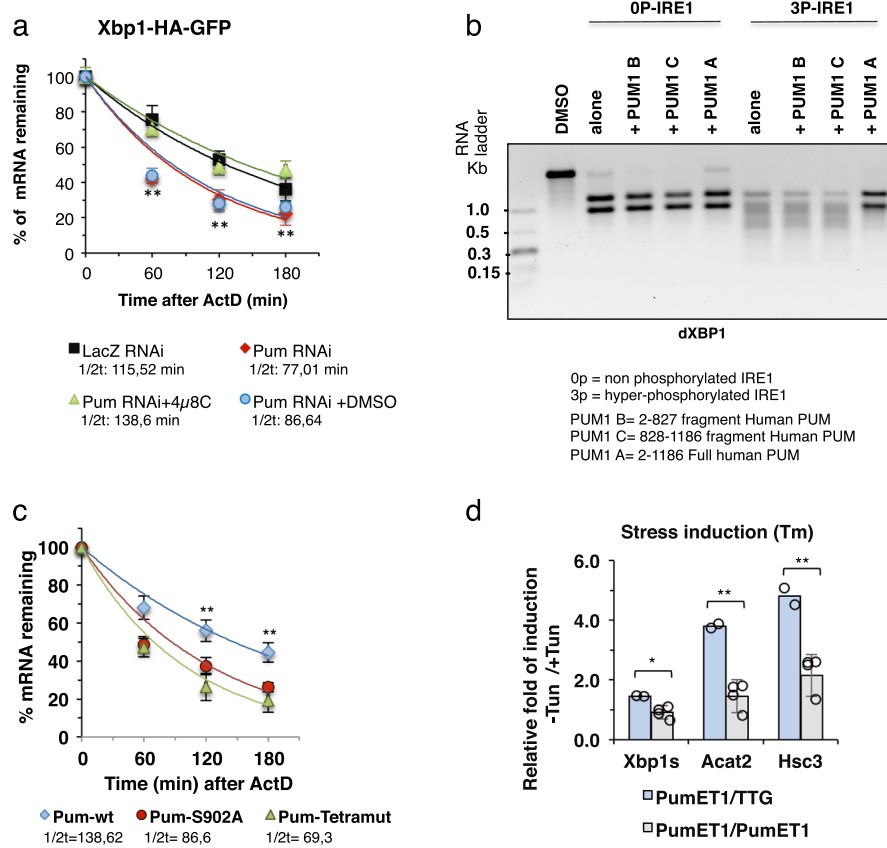

**Fig. 5 Pumilio protects Xbp1 mRNA from regulated Ire1-dependent decay. a** Control S2 cells (LacZ RNAi) were compared to cells treated with Pum RNAi and Pum RNAi treated with 4μ8C (100 μM), an inhibitor of IRE1 endoribonuclease activity, or vehicle control (DMSO). Total RNA was isolated after ER stress (5mM DTT) in the presence actinomycin D (5 μg/ml). qRT-PCR analysis was used to determine mRNA levels using specific primers for *gfp* and *rp49*. Results are presented as mean ± SD of %mRNA remaining. Half-lives were calculated by regression analysis. LacZ RNAi (black); PumRNAi (red); 4μ8C (green) and DMSO (blue). $n = 3$ biological independent experiments. one-way ANOVA coupled with Tukey's post hoc test (1 h: **$p = 0.0010053$ for lacZ RNAi, PumRNAi, PumRNAi DMSO, PumRNAi+4μ8C; 2 h: **$p = 0.0083911$ for LacZ RNAi vs PumRNAi; **$p = 0.0090392$ for PumRNAi vs PumRNAi +4μ8c. 3 h: *$p = 0.0183727$ for PumRNAi vs PumRNAi+4μ8C). **b** Human PUM1-FL protects Xbp1 RNA from IRE1 dependent non-canonical decay, but does not impair IRE1-dependent Xbp1 splicing. In vitro transcripts of dXbp1 were incubated with non-phosphorylated and phosphorylated forms of purified hIRE1α KR. dXbp1 = *Drosophila* Xbp1, hPUM1 B = (aa 2-827) of human PUM1; hPUM1 C = (aa 828-1186) of human PUM1, hPUM1 A = (aa 2-1186) of human PUM1; 0P-IRE1 - non-phosphorylated hIRE1α KR; 3P-IRE1 = phosphorylated hIRE1α KR. $n = 3$ of independent experiments. **c** qRT-PCR results from S2 cells transfected with full-length dPum (Pum-WT, blue), full-length dPum with a single phosphomutant (Pum-S902, red) or a quadruple phosphomutant (Pum-Tetramut - T537A, S540A, S544A, S902A, green). The Pumilio protective role upon Xbp1-HA-GFP is diminished in the dPUM-FL phosphomutants. One-way ANOVA coupled with Tukey's post hoc test (1 h: *$p = 0.0155$ for FL vs S902A and Tetra Mut; 2 h **$p = 0,0010$ for FL vs S902A,TetraMut). $n = 3$ of independent experiments. **d** RT-qPCRs for *Drosophila* Xbp1spliced and the Xbp1 targets Acat2 and Hsc3 (BiP) in PumET1 homozygous or control heterozygous PumET1/TTG larvae treated with/without tunicamycin (Tm) yeast paste food. The levels Xbp1spliced and the induction of Acat2 and Hsc3 upon tunicamycin treatment are diminished in PumET1 homozygous larva. Data are presented as mRNA levels relative fold induction (+/− Tm treatments): pumET1/TTG (blue bar, $n = 2$) and pumET1/PumET1(grey bar, $n = 4$). One-way ANOVA coupled with Tukey's post hoc test. Xbp1: *$p = 0.0174430$; Acat2: **$p = 0.0024205$; Hsc3: **$p = 0.0066179$. Error bars in (**a**, **c** and **d**) represent mean ± SD. Source data file is provided.

We do not know if Pumilio also protects Xbp1 mRNA from other ribonucleases or for how long the protective effect against RIDD/LE lasts. Other studies[61] have shown, using cultured mouse cells, an increase in the stability of Xbp1spliced mRNA at a time of translational repression during "early" phases of ER stress (4 h treatment with DTT). These authors suggested that Xbp1spliced mRNA could be protected from degradation by an unidentified protein factor. Our results suggest that Pumilio is such a factor, while Ire1 RIDD/LE activity is the degradation-causing agent.

From our model, we also predict that upon overcoming ER stress conditions, Pumilio phosphorylation levels will be reduced, either due to the activity of phosphatases and/or due to Ire1 dephosphorylation and attenuation[62]. At this stage, Pumilio's protective role over Xbp1 mRNA may subside, presumably making the transcript more vulnerable to the action of the mRNA

decay machinery and the ribosome-associated quality control, as previously shown for Hac1[63]. This regulatory step may avoid an accumulation of Xbp1spliced under non-stress conditions, which could be detrimental for cellular homeostasis.

Previous studies have shown that Pumilio proteins play a translational repressive role or promote degradation of their target mRNAs, which is in contrast with our results. However, another study has shown that, contrary to the canonical repressive activity, PUM1/2 rather promote FOXP1 expression through direct binding to two consensus PREs present in the FOXP1-3′ UTR[64]. Furthermore, an additional report has identified RNAs that are positively regulated by human PUM1 and PUM2[65].

Our results demonstrate that Ire1 is responsible for the phosphorylation status of Pumilio during ER stress and that this phosphorylation has implications for Xbp1spliced protein levels. Recent data in zebrafish implicate Pum1 phosphorylation as an

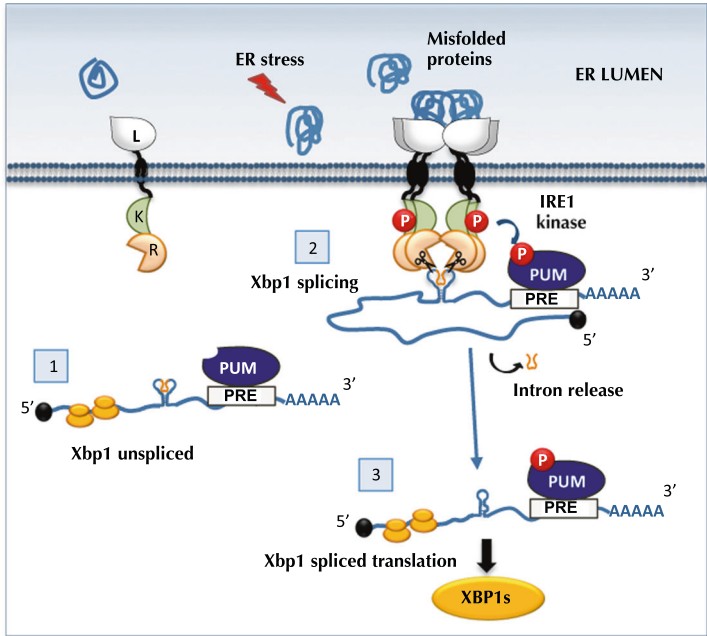

**Fig. 6 Model for Ire1-controlled post-transcriptional regulation of Xbp1 by Pumilio.** Step 1- In non ER stress conditions, Xbp1$^{unspliced}$ is present in the cytosol and Pumilio (PUM) may bind to the PRE elements in the Xbp1 3′UTR; Step 2- ER stress conditions lead to Ire1 autophosphorylation and PUM phosphorylation. Phosphorylated PUM protects Xbp1$^{spliced}$ mRNA from Ire1 dependent decay, but does not impair Ire1 dependent Xbp1 splicing. Step 3- Xbp1$^{spliced}$ transcripts can be efficiently translated into the Xbp1$^{spliced}$ transcription factor.

initial key step for the sequential activation of cyclinB1 mRNA translation during oocyte maturation, although the kinase involved remains unidentified[66]. In Xenopus, Nemo-like kinase (NLK)—a typical mitogen-activated protein kinase that is activated during an early phase of oocyte maturation[67]—was shown to directly phosphorylate Pum1 as well Pum2 in vitro. It is possible that other kinases may be involved in Pumilio phosphorylation under non-ER stress conditions or specific developmental stages.

In conclusion, our present work uncovers an unanticipated mechanism that regulates one of the key branches of the UPR through the action of an RNA binding protein. This involves Ire1-kinase-driven phosphorylation of Pumilio, which in turn protects Xbp1$^{spliced}$ mRNA from Ire1-RNase-driven RIDD and RIDDLE, thereby coordinating the two major endoribonuclease outputs of Ire1 to enable a more efficient intracellular response and ER stress mitigation.

## Methods

**Cell culture**. *Drosophila* S2 cells (Schneider, 1972) were cultured at 25 °C in Schneider's *Drosophila* medium (Invitrogen) supplemented with 10% heat inactivated fetal bovine serum (FBS, Invitrogen), 2 mM glutamine (Invitrogen), 100 U/mL penicillin and 100 μg/mL streptomycin (Invitrogen). MDA-MB-231 cells were all obtained from ATCC, authenticated by short tandem repeat (STR) profiles, and tested to ensure mycoplasma free within 3 months of use. All cell lines were cultured in RPMI1640 media supplemented with 10% (v/v) fetal bovine serum (FBS, Sigma), 2 mM glutaMAX (Gibco) and 100 U/ml penicillin plus 100 μg/ml streptomycin (Gibco).

**Drosophila stocks**. Transgenic lines were generated using pUAST-AttB and phiC31 integrase-mediated DNA integration (BestGene) that allows the insertion of the transgenes in a specific site of the acceptor fly genome. Clones of mutant eye tissue were generated by the Flp/FRT technique[68], with Flipase expression under the control of the *eyeless* promoter. Drosophila stocks obtained from the Bloomington Stock Center (Indiana University, Bloomington, IN, USA): GMR-Gal4 (active in the eye, under the control of the glass multiple reporter); eye-flip GMR-Gal4 (promotes recombination in the eye) and Actin5c- Gal4 (ubiquitous expression). *Drosophila* stocks were maintained at 25 °C on standard cornmeal media in an incubator with a 12 h light/dark cycle. TAP-PUM stocks were a kind gift from André Gerber. The stock *FRT, pum$^{ET1}$* was a kind gift from Yuh Nung Jan. Tunicamycin feeding experiments with Drosophila larva was done as

previously described[69]. Briefly, cages with apple juice plates (6 cm) were set with Pum$^{ET1}$/TM3 balancer with twiGAL4 > UAS-GFP) flies for 24 h and aged for 24 h more. At 48 h after egg laying, larva in the plates were exposed to 500 μL of yeast paste food with/without tunicamycin (5 mg/mL) for around 20 h, after which GFP positive (Control heterozygous) and GFP negative (Pum$^{ET1}$ homozygous) larva were manually collected under Leica fluorescent scope and processed for RNA extraction.

**Plasmid construction**. The 3′UTRs of Xbp1$^{spliced}$ and Xbp1$^{unspliced}$ forms were amplified from the cDNA clone GH09250 (Flybase) using specific primers (Supplementary Table 1) and cloned downstream of the green fluorescent protein (GFP) coding sequence in the vector pRmHa, containing the metallothionein promoter. The mutations in the Pumilio binding sites were introduced using oligonucleotide-mediated site-directed mutagenesis and inverse PCR. The UAS-Xbp1-HA-GFP construct was made by PCR cloning. All cloning was performed with the Phusion High-Fidelity PCR Master Mix with HF Buffer or GH Buffer according to the manufacturer's protocol. All clones were confirmed by sequencing (Stabvida). The full-length Pumilio protein with a C-terminal V5 tag[50] was a gift from Aaron Goldstrohm, from which the different truncated dPUM-D1, dPUM-D3 and dPUM-D1D3 were constructed.

**Transfection and stable cell line establishment**. Cells were co-transfected with the different plasmids using Effectene reagent according to manufacture indications (Qiagen). For stable transfection, cells were selected by replacing Schneider's complete *Drosophila* media with fresh medium supplemented with the appropriate antibiotics (zeocin, puromycin) according to the resistance gene present in the transfected plasmids and maintained under selective media until the formation of resistant clones. The reagents are described in Supplementary Table 2.

**dsRNA treatments**. RNAi was performed as described previously[70]. Primer pairs tailed with the T7 RNA polymerase promoter (Supplementary Table 1) were used to amplify PCR fragments obtained from cDNA clones. PCR products with an average size of 600 bp were then used as templates for dsRNA production with the T7 RiboMAX system (Promega). For transfection, 15 μg/ml dsRNA against *Drosophila* Pumilio or LacZ control were added to S2 cells in 12-well plates, during 9 days. mRNA depletion was confirmed by RT-PCR before further analysis.

**Total RNA and protein extraction**. For RNA stability and Western blot experiments, $4.5 \times 10^5$ S2 cells were seeded in 12-well plates the day before treatments, transfections and protein or RNA extraction. Transfection was performed with Effectene (Qiagen) according to the manufacturer's instructions. Expression of the GFP reporter under the metallothionein promoter was induced by adding 7 mM CuSO4 to the cell culture media for 3 h. For the UAS dependent Xbp1 reporters, transcription was induced by cotransfection of Actin-Gal4 plasmid. For mRNA

half-life measurements, transcription was blocked with actinomycin D (5 μg/ml; Sigma-Aldrich) and the cells were harvested at the indicated time points. Total RNA was then extracted using Zymo Research quick-RNA miniprep Kit.

**qPCR and RT-PCR**. Primers for qRT–PCR (Supplementary Table 1) were designed according to MIQE guidelines (Minimum Information for Publication of qRT-PCR Experiments - Supplementary Table 3) using NCBI primer blast, choosing a melting temperature of 62 °C. By using three serial dilutions of cDNA, primer efficiencies were determined, only primers with efficiencies varying around 100% were used for analysis.

Equal amounts of total RNA was retro-transcribed using RevertAid H Minus First Strand cDNA Synthesis Kit (Thermo/Fermentas). Each PCR reaction was performed on 1/40 of the cDNA obtained, using SSoFast EvaGreen Supermix (Bio-Rad) according to the manufacturer's instructions and Bio-Rad CFX-96 as detection system. All samples were analyzed in triplicates and from 3 independent biological RNA samples. For each sample, the levels of mRNAs were normalized using rp49 as a loading control. Normalized data then were used to quantify the relative levels of mRNA using the $\Delta\Delta CT$ method. qPCR was carried out using a CFX-96 Biorad instrument. Biological replicates represent independently grown and processed cells. Technical replicates represent multiple measurements of the same biological sample.

For experiments with human MDAMB231 cells, siRNAs treatments used either a non-targeting control (siNTC) (Dharmacon #D-001810-10-05) or PUM1 (#L-014179-00-0005) and PUM2 (#L-014031-02-0005) combined (siPUM1+2), following the RNAiMAX protocol from Invitrogen (#13778075). 72 h later, cells were treated for 6 h with DMSO or Tg, and then proceed to RNA extraction using the RNeasy Plus kit (Qiagen #74134). Equal amounts of RNA were reverse transcribed and amplified using the TaqMan™ RNA-to-CT™ 1-Step Kit (Applied Biosystems #4392938) on the ABI QuantStudio 7 Flex Real-Time PCR System. The delta-delta $C_T$ values were calculated by relating each individual $C_T$ value to its internal GAPDH control. Taqman primers for XBP1s (#Hs03929085_g1), SYVN1 (#Hs00381211_m1), PUM1 (#Hs00472881_m1), PUM2 (#Hs00209692_m1), and GAPDH (#Hs02758991_g1) were from Life Technology.

**Protein analysis**. Total protein lysates were prepared in lysis buffer containing protease inhibitors. Proteins were size-separated by SDS-PAGE and transferred onto nitrocellulose or PDVF membranes (Biorad). For Western blot analysis, primary antibodies (Supplementary Table 2) were rat anti-GFP (3H9) (1:1000, Chromotek), mouse anti-V5 (1:5000, Invitrogen), mouse anti-HA (1:5000, Covance) and mouse anti-α-tubulin (AA4.3) (1:1000, Developmental Studies Hybridoma Bank).

**Protein purifications**. Drosophila Pumilio domains (PUM-D1, PUM-D3, Pum-D1D3) fused to V5-6xHis were subcloned into pET 28a and pET26a vectors for protein expression in bacteria. Plasmids were transformed into BL21(DE3) competent cells and recombinant proteins were induced with 1 mM IPTG at 25 °C. Bacteria were lysed in lysis buffer (500 mM NaCl, 50 mM Tris-Cl, pH8.0, 0,1% Triton, Protease inhibitors). Samples were lysed by sonication, centrifuged twice at 14,000 × g for 15 min, and the cleared supernatant was bound to Ni-NTA Superflow beads (Qiagen) by gravity filtration. Unbound proteins were washed with lysis buffer at pH 8.0 supplemented with increasing amounts of imidazole (5 mM, 40 mM, 60 mM). Recombinant proteins were eluted from beads in lysis buffer with 100–250 mM imidazole at pH 8.0. For recombinant proteins retained in inclusion bodies (PUM-D3), solubilization was done by including 6M urea in lysis buffer. All purified Pumilio recombinant proteins were dialyzed overnight at 4 °C, against a final buffer (20 mM Hepes, 200 mM NaCl, 5% Glicerol, 2 mM DTT) and concentrated with appropriate MW cut-off Vivaspin columns (Merck). The concentration of purified proteins was determined by colorimetric assay (Bio-Rad DC Protein Assay) and verified by electrophoresis alongside of BSA standards with Coomassie staining.

**Immunoprecipitation of TAP-PUM from flies**. Extracts from adult flies heads were prepared as described in[48]. After immunoprecipitation of TAP-PUM, total RNA was extract with Trizol and Xbp1 mRNA levels detected by RT-PCR using specific primers for each Xbp1 transcript.

**Electrophoretic mobility shift assay**. EMSA was performed according to Chemiluminescent RNA EMSA Kit (Thermo scientific). RNA oligos (Supplementary Table 1) were designed to cover 30 bases of the Xbp1 3′UTR, including the Pumilio PRE, UGUACAUA sequence. The oligo pum1site-wt-3′BioTEG was labeled with biotin at the 3′ end, and the identical oligo without biotin labeling served as competitor probe. Mutant competitor oligo pum1site-mut contained the mutations UGUA to ACAA. For this experiment we used purified protein of human PUM1, containing the Pum-HD domain, which shares 80% of sequence conservation to Drosophila Pum-HD and both have similar affinity for Nanos response sequences[71].

Binding reactions were in a binding buffer containing 10 mM Hepes/KOH pH 7.4, 50 mM KCl, 3 mM MgCl₂, 1 mM EDTA, 0.1 mg/ml yeast tRNA, 2 mM dithiothreitol, 0.01% (w/v) Tween-20, 0.2 U rRNAsin (Promega), and 5% (v/v)

glycerol in a total volume of 20 μl (Chen et al., 2008). In the competition assay, the reaction mixture was supplemented and incubated with 5x, 10x and 50-fold molar excess of unlabeled competitor oligos before adding the biotin-labeled probe (10 nM). The protein-RNA complexes were allowed to form for 1 h on ice, followed by electrophoresis through 6% non-denaturing polyacrylamide gels in 0.5 X TBE at 100 V at room T °C. The samples were transferred to a Hybon N+ Membrane at 350 mA for 45′ at 4 °C, using Trans-Blot (Bio-Rad). The membrane was cross-linked for 3 min at 120.000 joules/cm2, followed by blocking for 1 hour in 5% milk in TBST (Tris-buffered saline, 0.1% Tween 20). Blots were incubated with HRP-conjugated anti-biotin antibody (1:1000, Cell Signaling, 7075P5) for 2 h at room temperature. After four times 5 min washing in TBST, the membrane was developed with ECL western blotting detection system (Amersham) and imaged using the ChemiDoc Imaging System (Bio-Rad).

**In vitro phosphorylation assays**. IRE1 phosphorylation assays were performed by incubation of purified hIRE1α KR (3,3 μg/μl) with purified Pumilio protein (dPumD1, dPumD3, hPumFl) in IRE1 kinase buffer (25 mM HEPES, pH 7.5, 150 mM NaCl, 5 mM DTT, 5% glycerol), containing either 10 μCi of γ-ATP[32P] for radioactive assays or cold ATP [2 mM] for phostag immunoblot assays (in a total volume of reaction of 20 μl). Inhibitors of IRE1 Kinase (Apy29 [2,5 μM/μl] and compound #18 [2,5 μM/μl]) were pre-incubated with hIRE1α KR and all reactions were assembled on ice, prior to addition of ATP and incubation for 2 h at 25 °C. Phosphatase treatment with λPP (NEB) or CIP (NEB) were performed after phosphorylation reactions, for 40 min at 30 °C. Each reaction was stopped by addition of SDS-PAGE loading buffer and run on pre-cast SDS-PAGE gels (Biorad). The autophosphorylation of hIRE1α KR was confirmed by western blot using a pSer phospho-specific antibody (Genentech) and phosphorylated Pumilio proteins were detected using anti-V5 antibody (Invitrogen). In the case of kinase radioactive assays, gels were dried and kinase activity visualized by autoradiography.

**Phostag-gels**. S2 cells were lysed in CIP buffer (100 mM NaCl, 50 mM Tris- HCl pH 7.9, 10 mM MgCl, 1 mM DTT), 1 mM PMSF, 0.1% NP40, protease inhibitor cocktail (Roche), and phosphatase inhibitor cocktail (Calbiochem). For CIP treatment phosphatase inhibitor cocktail was omitted, and lysate was incubated at 37 °C, for 60 min in 2 units CIP (NEB) per 50 μL reaction containing 50μg of total protein. For λ-phosphatase (NEB) treatment, lysates were incubated with 1 unit of phosphatase for 30 min at 30 °C in λ−phosphatase buffer. Lysates were cleared by centrifugation and subjected to SDS-PAGE. To detect phosphorylated Pumilio in SDS-PAGE, we used Phos-tag AAL-107 (Wako Chemicals GmbH) according to the manufacturer's instruction[72,73]. Western blotting was performed using mouse anti-V5 (1:5000, Invitrogen), followed by the corresponding Horseradish Peroxidase (HRP) conjugated secondary antibodies (1:5000, GE Healthcare) and visualized using the ECL Plus Western Blotting detection system (GE Healthcare).

**Immunofluorescence and confocal microscopy**. For Drosophila pupal dissections, white pre-pupae (0h pupa) were collected and maintained at 25 °C until the required stage. Larval, pupal and adult eyes were dissected in 1xPBS, fixed in 1xPBS + 4% Formaldehyde for 40 min at room temperature and washed 3 times with 1xPBS  0.3% Triton X- 100. Primary antibodies (Supplementary Table 2) were incubated in 1xPBS, 1% BSA, 0.1% Tween 20, 250 mM NaCl overnight at 4 °C. Samples were washed 3 times with 1xPBS + 0.3% Triton X-100 and incubated with appropriate secondary antibodies (from Jackson Immuno- Research Laboratories) for 2 h at room temperature. Samples were mounted in 80% glycerol in a bridge formed by two cover slips to prevent the samples from being crushed while analyzed on the confocal microscope (Leica TCS SP5, 63X magnification oil immersion lens).

**Xbp1 RNA cleavage assay**. T7 RNA was generated from pOT2-Xbp1, containing the ORF and UTR of Drosophila Xbp1. 1 μg of RNA was digested at room temperature by 1 μg of human IRE1α KR 0P or 3P recombinant protein (~0.8 μM) for 45 min in RNA cleavage buffer (HEPES pH7.5 20 mM; K acetate 50 mM; Mg acetate 1 mM; TritonX-100 0.05% (v/v)). The total volume of the reaction was 25 μl. The digestion was then complemented by an equal volume of formamide and heated up at 70 °C for 10 min to denature the RNA. The mixture was immediately placed on ice for 5 min, and then 20 μl was run on 3% agarose gel at 160 V for 1 h at 4 °C. The PUM proteins were incubated with the RNA for 40 min on ice prior to RNA digestion.

**Statistical methods**. All panels data are represented as mean ± SD, from at least three independent biological replicates experiments. All statistical comparisons for significance between control and experimental groups was calculated using a significance cut off $p < 0.05$. and denoted by *$p < 0.05$,**$p < 0.01$, and ***$p < 0.001$, based on two-tailed unpaired t-Student's test or one-way ANOVA followed by an appropriate post-hoc analysis. Statistical analyses were performed using GraphPad Prism 8 (GraphPad Software, Inc) and online resources (https://astatsa.com/OneWay_Anova_with_TukeyHSD).

**Reporting summary**. Further information on research design is available in the Nature Research Reporting Summary linked to this article.

## Data availability

The data supporting the findings of this study are available from the corresponding authors upon reasonable request. Source data for each figure are provided with this paper as a source data file. Source data are provided with this paper.

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

## Acknowledgements

We thank Aaron Goldstrohm for plasmids. We thank Yuh Nung Jan, Stefan Luschnig and André Gerber for Pumilio fly stocks. We thank David Ron and Heather Harding for providing the purified hIRE1 ⟨ KEN, 4µ8C and helpful discussions. This work received funding by grants from the Fundação para a Ciencia e Tecnologia (PTDC/BIA-BCM/105217/2008; PTDC/SAU- OBD/104399/2008; PTDC/BEX-BCM/1217/2012; FCT-ANR/NEU-NMC/0006/2013; PTDC/NEU-NMC/2459/2014; SFRH/ BPD/93893/2013 and DL57/2016 to FC and SFRH/BD/ 130817/2017 to CS). The project leading to these results has also received funding from 'la Caixa' Foundation (ID 100010434), under the agreement <LCF/PR/HR17/52150018>.

## Author contributions

F.C. designed the study, performed the experiments, analyzed data and wrote the manuscript. C.S. constructed reagents for the Phostag assays and contributed to the RNA IP assays. A.L.T. conducted the Xbp1 mRNA in vitro cleavage assays and experiments with human cells. S.M. provided the purified versions of hPUM1. P.M.D. performed the IF experiments in Fig. 2. P.M.D. and A.A. designed and supervised the study, analyzed data and wrote the manuscript. All authors read and edited the manuscript.

## Competing interests

A.LeT., S.M., and A.A. were employees of Genentech, Inc., a member of the Roche Group, during the performance of these studies. All other authors declare no competing interests.
