## [Peer Review File · Nature Communications]

Pumilio protects Xbp1 mRNA from regulated Ire1-dependent decayReviewers' Comments:

Reviewer #1:

Remarks to the Author:

The manuscript by Carriao et al investigated the role of the RNA binding protein, Pumilio, in the protection of Xbp1 mRNA against IRE1's RIDD activity. The authors present several lines of experimental evidence to support their claims. They use an Xbp1-GFP reporter construct to show that Pumilio can protect *Drosophila* Xbp1 mRNA from IRE1's RIDD activity. Moreover, the authors also show that this protection is dependent on the phosphorylation of Pumilio by IRE1.

I would like to congratulate the authors on this very interesting observation and their work which contains some excellent biochemistry. However, I have some points aimed at improving the manuscript. In particular, the manuscript does not fully articulate the major observation. Also, further clarity and detailed explanation of experimental procedures in the main text and figure legends is required to make the manuscript easier for the general readership to follow. This is particularly evident in Figures 2 and 5 where the main text and the figure don't seem to be consistent.

As such, I recommend the following to help the authors improve their manuscript.

Major issues:

1. It seems to me that the most exciting and important finding of this manuscript is somewhat hidden for a non-specialist reader. When IRE1 cleaves the dual hairpin loops of XBP1 mRNA, the fragments that result are re-ligated, producing a single hairpin loop that is also a potential substrate for IRE1. However, it has never been clear why XBP1s is not in turn cleaved by IRE1 as are other RIDD substrates. The manuscript provides an explanation. However, I think that the impact of the work could be greater if this was articulated better in the Abstract, Introduction and Discussion sections. Also perhaps a change to Fig6 to reflect the single stem loop in XBP1s mRNA – Please see the Fig1a in the manuscript by Jurkin et al EMBO J (2014)33:2922-2936.

2. The authors have not demonstrated that Pumilio regulated Xbp1 mRNA through binding to its PER domains or even to the 3'UTR region, rather they have implied this. Given the significance of their work this would be an important (and relatively straightforward) experiment to include here.

3. Figure 2a: I had difficulty understanding the authors' presentation of their Xbp1 construct. While the outcomes of Xbp1 splicing on their construct are clear, the figure itself is confusing. The authors start from mRNA and seem to go directly to the protein without any description or labelling of the figure. Splicing is shown to cause complete removal of both stem loops, rather than cleavage of each stem loop and re-ligation of the fragments. The 3'UTR portion is lacking and the GFP sequence disappears in unstimulated conditions. It's not clear that the upper part of the figure is mRNA and the lower part is protein. The depiction of in-frame and out-of-frame sequences is not consistent. In my opinion, the authors should stay at mRNA level and add stripes on out-of-frame portions (such as GFP in unspliced XBP1) as they did for HR2 and HA in spliced Xbp1. If the authors prefer to show the final product of their construct, please add description that what they show is the protein to avoid confusion. Second, the "In" box is not described in the figure legend. There needs to be more explanation of the construct either in the figure or the legend.

4. Figure 2B: The Western Blot has no loading control. The WT and Pum-V5 overexpressing should be from the same experiment and run on the same blot

5. Figure 2C: On page 3, the text referring to figure 2C talks only of the stability of the Xbp1-HA-GFP reporter mRNA. In this paragraph, the authors don't differentiate between spliced and unspliced version concerning the measurement of their construct mRNA stability which makes sense to me as the figure 2C seems to show that the RT-qPCR has been done on GFP sequences as performed in Figure 1. Indeed, while spliced and unspliced version will translate into GFP or HA carrying forms of

XBP1 protein, both spliced and unspliced versions of the mRNA should have the GFP sequence. Could the authors clarify which primers they used to enable them to distinguish between spliced and unspliced XBP1 construct in this experiment? Could the authors explain why figure 2C title and legend says that it represents "Xbp1-HA-GFP:3'UTR spliced" version and why total RNA was extracted after ER stress induction by DTT?

6. Fig 2C: There are no controls to show that Pumilio has been knocked down or overexpressed (PumHD or Full). How do the overexpression constructs avoid RNAi downregulation?

7. Figure 3D: Could the authors explain the presence of the higher band in the CIP column? Isn't CIP supposed to remove or highly diminish the presence of this phosphorylated form of D1? Moreover, a better alpha-tubulin blot is required as the protein is missing in one of the lanes. Could the authors also label the second column of this picture?

8. Page 6, end of the first section: Could the authors explain why they tested specifically the S902 phosphorylation site alone and not the others?

9. Figure S2a: the authors need to define what are PMCa, PMCb and PUF

10. Figure S2b: Panel hPUM1A Lane 5, do the authors have an explanation to the presence of such a high signal of phospho-IRE1 in presence of lambdaPP? Figure 4a and b seem to show that lambdaPP is highly potent to reduce phospho-IRE1alpha signalling. While we can see a lower signal that in lane 4, it seems very high compared to the other conditions with IRE1 kinase inhibition. The same question applies to the last column of the hPUM1B panel and fifth column of the hPUM1C panel.

11. Although the authors have demonstrated that human IRE1 can phosphorylate Pumilio and that it can phosphorylate human Pum1, the more interesting question is whether human Pum1 can regulate human/mammalian XBP1s mRNA. The authors should show the regulation of human XBP1s mRNA by human Pum1.

12. Statistical analysis: Many figure legends state that the data are expressed as means and standard deviation while the material and methods says that all panels data are represented as mean and SEM. N values should be included for all experiments.

13. It would be extremely interesting to know if there is a similar mechanism operating in mammalian cells to protect XBP1s mRNA from RIDD or from degradation that is not dependent on endonucleases. Could the authors speculate on this in the discussion?

Minor points:

1. Page 2: First line, second column, "To test experimentally if these regions can regulate mRNA". Please specify what feature of mRNA is concerned.

2. Page 3, Figure 1b: Please reverse the order of the two panels in order to keep a natural reading order/direction.

3. Page 3, Figure 1c: Could the authors put a legend on the * symbol to say what it is supposed to be?

4. Page 3, end of the page: There a typo on Xbp1-HA-GFP.

5. Page 6: "D1= aa 1 to 547; D3 = aa 77 to 1091" it seems that the position of the beginning of D3 is incorrect. Please check.

6. Page 6: "relative to non-ER stress conditions (0 h DTT) (Fig.3b)." Please add also figure 3c.

7. Figure S1: Please stay consistent with Figure 1B presentation. Please also add the SD to this figure.

8. The manuscript needs to be proofed for English and typographical errors. Clearer and more detailed descriptions in figure legends are needed (it is frequently unclear what are the treatments, or which primers/antibodies were used).

Reviewer #2:

Remarks to the Author:

Work presented by Cairrao and colleagues examines the role of the PUMILIO RNA-binding protein in binding to and regulating the RNA stability of Xbp1 RNA isoforms in *Drosophila*. This interesting manuscript identifies a unique role for PUMILIO in helping to stabilize the Xbp1 RNA spliced isoform following stress and details the molecular mechanisms that links stress to PUMILIO regulation. Despite my appreciation for this work, there are some significant control gaps and image quality issues that require remediation before this work can contribute to the field.

Major:

1. Untransfected control cells are not the right control for siRNA or dsRNA experiments. The correct control should be cells transfected with a Scrambled or like siRNA/dsRNA as transfection and exogenous RNAs do significantly alter cellular processes. Currently all of the RNAi figures in this manuscript use untransfected cells as a "control".

2. Low quality data presentation also makes the visual interpretation of some of the data difficult. In particular, the data presented in Figure 2B, 2D and 4A, is poor. I think that underlying data is interesting and correctly interpreted but requires improvement. For Figure 2B, I would like to see a quantification of changes and the use of an additional PUM over-expression control. Ideally, this control would have mutations that diminished PUM binding.

3. In Figure 5A, there are no controls, for either siRNA treatment or 4 μ 8c treatment. Both need control siRNAs and vehicle (for 4 μ 8c) controls. In addition, 4 μ 8c should be used alone.

4. In Figure 2 and Figure 5, RNAi and over-expression experiments are conducted and interpreted without evidence that these treatments worked. This limits my enthusiasm for this data.

5. The authors provide no evidence that Pumilio regulation of XBP1 is important for cellular response to stress. This speaks to the importance of this mechanisms in the stress response and needs significant expansion.

Minor:

1. Having single and double PRE mutants presented in both the spliced and unspliced data presented in Figure 1B and S1 would improve the analysis.

2. Please define: * in Figure 1C.

3. The rationale behind investigating phosphorylation of PUMILIO is not addressed and although the data supports this, smoothing this transition in the text would help the reader.

4. Figure 5B states "since we could not purify full-length Pumilio..", however the next lines and figure discuss "full-length Pumilio", could the authors please clarify/correct.

5. The "tetraMUT" also comes from nowhere. I think its interesting data, but please expand its description before use.

6. The discussion is nicely constructed however a greater inclusion of the phosphorylation data and its implications for Pumilio in general would help to expand the utility of these findings.

Reviewer #3:

Remarks to the Author:

The manuscript entitled "Pumilio protects Xbp1 mRNA from regulated Ire1-dependent decay" by Carriao and colleagues studied the functions of Pumilio in regulating the splicing of Xbp1 mRNA, which is important in ER stress. They conducted biochemical analyses and *Drosophila* models in this study. They found that Pum interacts with 3'UTR of fly Xbp1 mRNA and can be phosphorylated by kinase Ire1.

This study is well designed and performed. However, since most experiments are in the fly system, I

feel it is quite preliminary to publish in Nature Communication for now. Generally, I expect the authors could validate their findings in human or other mammalian models.

Besides this major suggestion, I also have a few minor comments:

1. Figure 1c, missing an immunoblot showing the protein pull-down. Also why Xbp1 mRNA can also be purified from the control group. The specificity of RNA-IP is questionable.
2. Figure 2c, are the PumHD and FullPum constructs used in this assay RNAi resistant?
3. What are the subcellular localization of Pum and Ire1 kinase, in the cytosol or in ER?
4. The authors claimed that "Indeed, Le Thomas et al showed that in its purified, fully phosphorylated oligomeric form hIRE1 α KR (KR-3P) degraded not only RIDD and RIDDLE mRNA targets, but also Xbp1 mRNA." And the made hypothesis on it. However, there is no data or reference supporting this notion.

We would like to thank the 3 reviewers for the global positive evaluation of our manuscript, and specially for the many comments and revision requests, that, we feel, have much improved this revised version. We detail below our actions (in blue font) in response to each of the reviewers comments (italic). In the text file, we also highlighted in blue font all the changes we made to the manuscript. We hope you find this revised version suitable for publication in *Nature Communications*.

Reviewer #1 (Remarks to the Author):

The manuscript by Carriao et al investigated the role of the RNA binding protein, Pumilio, in the protection of Xbp1 mRNA against IRE1's RIDD activity. The authors present several lines of experimental evidence to support their claims. They use an Xbp1-GFP reporter construct to show that Pumilio can protect Drosophila Xbp1 mRNA from IRE1's RIDD activity . Moreover, the authors also show that this protection is dependent on the phosphorylation of Pumilio by IRE1.

I would like to congratulate the authors on this very interesting observation and their work which contains some excellent biochemistry. However, I have some points aimed at improving the manuscript. In particular, the manuscript does not fully articulate the major observation. Also, further clarity and detailed explanation of experimental procedures in the main text and figure legends is required to make the manuscript easier for the general readership to follow. This is particularly evident in Figures 2 and 5 where the main text and the figure don't seem to be consistent.

As such, I recommend the following to help the authors improve their manuscript.

Major issues:

1. It seems to me that the most exciting and important finding of this manuscript is somewhat hidden for a non-specialist reader. When IRE1 cleaves the dual hairpin loops of XBP1 mRNA, the fragments that result are re-ligated, producing a single hairpin loop that is also a potential substrate for IRE1. However, it has never been clear why XBP1s is not in turn cleaved by IRE1 as are other RIDD substrates. The manuscript provides an explanation. However, I think that the impact of the work could be greater if this was articulated better in the Abstract, Introduction and Discussion sections. Also perhaps a change to Fig6 to reflect the single stem loop in XBP1s mRNA – Please see the Fig1a in the manuscript by Jurkin et al EMBO J (2014)33:2922-2936.

We have made alterations to the text that hopefully will convey better the message of the manuscript. We have made the correction to Fig.6, as suggested.

2. *The authors have not demonstrated that Pumilio regulated Xbp1 mRNA through binding to its PER domains or even to the 3'UTR region, rather they have implied this. Given the significance of their work this would be an important (and relatively straightforward) experiment to include here.*

We now have in Fig. 1c an EMSA experiment where we show that Pum-HD directly binds the PRE1 found in Xbp1 3'UTR.

3. *Figure 2a: I had difficulty understanding the authors' presentation of their Xbp1 construct. While the outcomes of Xbp1 splicing on their construct are clear, the figure itself is confusing. The authors start from mRNA and seem to go directly to the protein without any description or labelling of the figure. Splicing is shown to cause complete removal of both stem loops, rather than cleavage of each stem loop and re-ligation of the fragments. The 3'UTR portion is lacking and the GFP sequence disappears in unstimulated conditions. It's not clear that the upper part of the figure is mRNA and the lower part is protein. The depiction of in-frame and out-of-frame sequences is not consistent. In my opinion, the authors should stay at mRNA level and add stripes on out-of-frame portions (such as GFP in unspliced XBP1) as they did for HR2 and HA in spliced Xbp1. If the authors prefer to show the final product of their construct, please add description that what they show is the protein to avoid confusion. Second, the "In" box is not described in the figure legend. There needs to be more explanation of the construct either in the figure or the legend.*

We made alterations to Fig. 2a to take in consideration the points raised by the reviewer. We now indicate with labels in the figure the reporter at the "mRNA" level and the "Protein" end products that result from the presence or absence of ER stress. We also made a change to indicate the partial removal of the hairpin structure by Ire1. We also made changes to the figure legend.

4. *Figure 2B: The Western Blot has no loading control. The WT and Pum-V5 overexpressing should be from the same experiment and run on the same blot* We have made these corrections in Fig. 2b. We also expanded Fig. 2b with the use of PumR7 mutant (reduced mRNA binding), as requested by Reviewer 2. Blots shown are from same experiment/same blot/same exposure (for each antibody). Break line before PumR7 is just because the order of the PumR7 samples was inverted in the gel.

5. *Figure 2C: On page 3, the text referring to figure 2C talks only of the stability of the Xbp1-HA-GFP reporter mRNA. In this paragraph, the authors don't differentiate between spliced and unspliced version concerning the measurement of their construct mRNA stability which makes sense to me as the figure 2C seems to show that the RT-qPCR has been done on GFP sequences as performed in Figure 1. Indeed, while spliced and unspliced version will translate into GFP or HA carrying forms of XBP1 protein, both spliced and unspliced*

versions of the mRNA should have the GFP sequence. Could the authors clarify which primers they used to enable them to distinguish between spliced and unspliced XBP1 construct in this experiment? Could the authors explain why figure 2C title and legend says that it represents “Xbp1-HA-GFP:3’UTR spliced” version and why total RNA was extracted after ER stress induction by DTT?

The experiment in Fig. 2C was done with primers for GFP, so we are looking at the “total” mRNA levels of Xbp1-HA-GFP. In this case we are not using the primers that distinguish Xbp1spliced from Xbp1unspliced. However, because the experiment was done after treating the cells with DTT for 4 hours, almost all Xbp1 mRNA will be in the spliced form, certainly. We have changed the title in the figure from “Xbp1-HA-GFP:3’UTR spliced” to “Xbp1-HA-GFP”. We agree this was confusing. The Xbp1-HA-GFP reporter was done using the 300bp 3’UTR of Xbp1spliced, and the title was trying to convey this fact, but we now explain this in Fig 2a legend, instead.

6. Fig 2C: There are no controls to show that Pumilio has been knocked down or overexpressed (PumHD or Full). How do the overexpression constructs avoid RNAi downregulation?

We now show RT-PCR results (Supplementary Fig 1c) showing the specific and efficient knock down of Pumilio in the S2 cells PumRNAi treatments. For the overexpression constructs we used plasmids that lack the Pum 3’UTR region that is the targeted by Pum RNAi.

7. Figure 3D: Could the authors explain the presence of the higher band in the CIP column? Isn’t CIP supposed to remove or highly diminish the presence of this phosphorylated form of D1? Moreover, a better alpha-tubulin blot is required as the protein is missing in one of the lanes. Could the authors also label the second column of this picture?

We modified Fig. 3d and we now present novel panels for tubulin (with the correct number of lanes). We also added a triple phospho mutant (M3 = T537A, S540A, S544A) that shows reduced levels of D1 phosphorylation. We agree there is some minor residual phosphor bands in the CIP treated lane, in the panel where we also do 4u8C treatments. However, we do not think this affects the significance of the result which is that inhibition of Ire1 RNase activity (by 4u8C treatment) does not diminish PumD1 phospho bands. Hence, Pum D1 phosphorylation requires Ire1 kinase activity, but not Ire1 RNase activity.

8. Page 6, end of the first section: Could the authors explain why they tested specifically the S902 phosphorylation site alone and not the others?

For the PhosTag experiments we had to make 3 different constructs (D1, D3, D1-D3, which we describe in the manuscript), because the full length Pumilio protein (around 180 KDa) was too big to cause nice band migration differences in the Phostag gels. So in Fig 3, we analyzed S902 in the context of D3 and (now) the M3 triple mutant (T537A, S540A, S544A) in the context of D1.

9. *Figure S2a: the authors need to define what are PMCa, PMCb and PUF*
PCMa and PCMb refer to the “Pumilio Conserved Motif” a and b, domains of conservation in the Pumilio proteins of several species, including *Drosophila* and humans, as defined in references 36, 42, 50. We make note of this in the text. We have substituted PUF for PUM HD (Pumilio Homology Domain) and this is the domain of Pumilio that directly binds to mRNAs, as we also describe.

10. *Figure S2b: Panel hPUM1A Lane 5, do the authors have an explanation to the presence of such a high signal of phospho-IRE1 in presence of lambdaPP? Figure 4a and b seem to show that lambdaPP is highly potent to reduce phospho-IRE1alpha signalling. While we can see a lower signal that in lane 4, it seems very high compared to the other conditions with IRE1 kinase inhibition. The same question applies to the last column of the hPUM1B panel and fifth column of the hPUM1C panel.*

We agree with the reviewer that Supplementary 2b: Panel hPUM1A Lane 5 presents significant levels of phospho-IRE1 despite lambdaPP treatment. The only explanation we have is that perhaps lambdaPP activity was reduced in this specific experiment. Nevertheless, we still have strong reduction of phospho-IRE1 with the APY29 and compound#18 kinase inhibitors, so we think our conclusions still stand.

11. *Although the authors have demonstrated that human IRE1 can phosphorylate Pumilio and that it can phosphorylate human Pum1, the more interesting question is whether human Pum1 can regulate human/mammalian XBP1s mRNA. The authors should show the regulation of human XBP1s mRNA by human Pum1.*

We now present results in Supplementary Fig. 4a, showing that human PUM1 can protect human XBP1 from non-canonical IRE1 dependent mRNA decay. We also show in Sup Fig 4b that siRNAs for PUM1 and PUM2 in human MDAMB231 cells lead to a reduction in the levels of hXBP1 and the XBP1 target gene SYVN1, upon treatment with Thapsigargin, to induce ER stress. These results are similar to the results we obtained for *Drosophila* Xbp1 (Fig. 5).

12. *Statistical analysis: Many figure legends state that the data are expressed as means and standard deviation while the material and methods says that all panels data are represented as mean and SEM. N values should be included for all experiments.*

We have corrected these issues and also present Tukey’s test for statistical significance of the results, where appropriate.

13. *It would be extremely interesting to know if there is a similar mechanism operating in mammalian cells to protect XBP1s mRNA from RIDD or from degradation that is not dependent on endomotifs. Could the authors speculate on this in the discussion?*

We now present results in Sup. Fig 4a, showing that human PUM1 can protect human XBP1 from non-canonical IRE1 dependent mRNA decay. We also show in Sup Fig 4b that siRNAs for PUM1 and PUM2 in human MDAMB231 cells lead to a reduction in the levels of hXBP1 and the XBP1 target gene SYVN1, upon treatment with Thapsigargin, to induce ER stress. These results are similar to the results we obtained for *Drosophila* Xbp1 (Fig. 5).

Minor points:

1. Page 2: First line, second column, "To test experimentally if these regions can regulate mRNA". Please specify what feature of mRNA is concerned.

We corrected this sentence to "To test experimentally if these putative PREs can regulate mRNA,"

2. Page 3, Figure 1b: Please reverse the order of the two panels in order to keep a natural reading order/direction.

We have made this correction.

3. Page 3, Figure 1c: Could the authors put a legend on the * symbol to say what it is supposed to be?

This is now not relevant as we substituted the data in Fig. 1c with the EMSA panel, as suggested by the reviewer.

4. Page 3, end of the page: There a typo on Xbp1-HA-GPF.

Corrected to Xbp1-HA-GFP

5. Page 6: "D1= aa 1 to 547; D3 = aa 77 to 1091" it seems that the position of the beginning of D3 is incorrect. Please check.

Yes, corrected to aa 777.

6. Page 6: "relative to non-ER stress conditions (0 h DTT) (Fig.3b)." Please add also figure 3c.

Corrected to "relative to non-ER stress conditions (0 h DTT) (Fig.3b, c)."

7. Figure S1: Please stay consistent with Figure 1B presentation. Please also add the SD to this figure.

We added SD/error bars to the figure.

8. The manuscript needs to be proofed for English and typographical errors. Clearer and more detailed descriptions in figure legends are needed (it is frequently unclear what are the treatments, or which primers/antibodies were used).

We hope the improvements we made are satisfactory.

Reviewer #2 (Remarks to the Author):

Work presented by Cairrao and colleagues examines the role of the PUMILIO RNA-binding protein in binding to and regulating the RNA stability of Xbp1 RNA isoforms in Drosophila. This interesting manuscript identifies a unique role for PUMILIO in helping to stabilize the Xbp1 RNA spliced isoform following stress and details the molecular mechanisms that links stress to PUMILIO regulation. Despite my appreciation for this work, there are some significant control gaps and image quality issues that require remediation before this work can contribute to the field.

Major:

1. Untransfected control cells are not the right control for siRNA or dsRNA experiments. The correct control should be cells transfected with a Scrambled or like siRNA/dsRNA as transfection and exogenous RNAs do significantly alter cellular processes. Currently all of the RNAi figures in this manuscript use untransfected cells as a “control”.

We now use RNAi against LacZ as control in all dsRNA experiments, instead of untransfected cells.

2. Low quality data presentation also makes the visual interpretation of some of the data difficult. In particular, the data presented in Figure 2B, 2D and 4A, is poor. I think that underlying data is interesting and correctly interpreted but requires improvement. For Figure 2B, I would like to see a quantification of changes and the use of an additional PUM over-expression control. Ideally, this control would have mutations that diminished PUM binding.

We now present novel, hopefully more convincing, figures for Fig 2b and 2d. For Fig. 2b we show quantification of the changes. Also in Fig2b, we now show results with the Pumilio R7 mutant (with diminished RNA binding – obtained from the Goldstrohm lab), that did not cause any change in the levels of Xbp1^{spliced} or Xbp1^{unspliced}, in comparison with controls.

3. In Figure 5A, there are no controls, for either siRNA treatment or 4μ8c treatment. Both need control siRNAs and vehicle (for 4μ8c) controls. In addition, 4μ8c should be used alone.

We now present a revised version of this figure where we show LacZ RNAi control and vehicle (DMSO) control for 4μ8C treatment.

4. In Figure 2 and Figure 5, RNAi and over-expression experiments are conducted and interpreted without evidence that these treatments worked. This limits my enthusiasm for this data.

We now show RT-PCR results (Supplementary Fig 1c) showing the specific and efficient knock down of Pumilio in the S2 cells PumRNAi treatments related to Fig. 2 and Fig. 5.

5. The authors provide no evidence that Pumilio regulation of XBP1 is important

for cellular response to stress. This speaks to the importance of this mechanisms in the stress response and needs significant expansion.

We address this point in Fig 5d, by showing diminished ER stress response (diminished Xbp1s levels and diminished levels of Xbp1 targets Acat2 and HSC3) in *Drosophila* larva homozygous for the Pumilio mutation (Pum^{ET1}), upon Tunicamycin treatments. We also show similar diminish ER stress response (Supplementary Fig. 4b) in experiments using human MDAMB231 cells treated with Thapsigargin (Tg) and siRNAs for PUM1 and PUM2 (siPUM1+2).

Minor:

1. Having single and double PRE mutants presented in both the spliced and unspliced data presented in Figure 1B and S1 would improve the analysis. We now present results with single and double PRE1/PRE2 mutants.

*2. Please define: * in Figure 1C.*

This is no longer relevant as we now have a different experiment in Fig. 1c - an EMSA experiment where we show that Pum-HD directly binds the PRE1 found in Xbp1 3'UTR.

3. The rationale behind investigating phosphorylation of PUMILIO is not addressed and although the data supports this, smoothing this transition in the text would help the reader.

We make a reference in the text "It is known that Pumilio proteins may become activated to regulate target mRNAs upon phosphorylation⁵³". Reference 53 – Kedde et al, 2010 – is a study where it was shown that PUM1 is activated by phosphorylation to regulate the p27 mRNA. This was the rationale for us to investigate phosphorylation of dPum during ER stress.

4. Figure 5B states "since we could not purify full-length Pumilio..", however the next lines and figure discuss "full-length Pumilio", could the authors please clarify/correct.

We could not purify *Drosophila* full-length Pumilio, but we could purify and use in our experiments full-length human Pum1. We corrected this in the text.

5. The "tetraMUT" also comes from nowhere. I think its interesting data, but please expand its description before use.

We now present results in Fig 3, where we show the single phosphomutant S902A in the context of D3 and the M3 triple mutant (T537A, S540A, S544A) in the context of D1. For these PhosTag experiments we had to make 3 different constructs (D1, D3, D1-D3, which we describe in the manuscript), because the full length Pumilio protein (around 180 KDa) was too big to cause nice band migration differences in the Phostag gels. So in Fig 5C, we analyzed S902A and the tetramutant (T537A, S540A, S544A, S902A) in the context of the full length protein. We made corrections to the text that make this clear, we hope.

6. The discussion is nicely constructed however a greater inclusion of the

phosphorylation data and its implications for Pumilio in general would help to expand the utility of these findings.

We corrected this in the discussion.

Reviewer #3 (Remarks to the Author):

The manuscript entitled "Pumilio protects Xbp1 mRNA from regulated Ire1-dependent decay" by Carriao and colleagues studied the functions of Pumilio in regulating the splicing of Xbp1 mRNA, which is important in ER stress. They conducted biochemical analyses and Drosophila models in this study. They found that Pum interacts with 3'UTR of fly Xbp1 mRNA and can be phosphorylated by kinase Ire1.

This study is well designed and performed. However, since most experiments are in the fly system, I feel it is quite preliminary to publish in Nature Communication for now. Generally, I expect the authors could validate their findings in human or other mammalian models.

We now present results in Sup. Fig 4a, showing that human PUM1 can protect human XBP1 from non-canonical IRE1 dependent mRNA decay. We also show in Sup Fig 4b that siRNAs for PUM1 and PUM2 in human MDAMB231 cells lead to a reduction in the levels of hXBP1 and the XBP1 target gene SYVN1, upon treatment with Thapsigargin, to induce ER stress. These results are similar to the results we obtained for *Drosophila* Xbp1 (Fig. 5).

Besides this major suggestion, I also have a few minor comments:

1. *Figure 1c, missing an immunoblot showing the protein pull-down. Also why Xbp1 mRNA can also be purified from the control group. The specificity of RNA-IP is questionable.*

We now show an immunoblot with PumHD inputs and pull downs (Sup .Fig 1C). Also we note that the PumHD pull down methodology was developed and validated by Gerber et al 2006 (Ref. 48), where they did the genome-wide identification of mRNAs that associate with PumHD. In fact, in the Sup. Table 2 of Gerber et al, Xbp1 is listed among the hundreds of mRNAs that were identified as being pulled down by PumHD. In our manuscript, the goal was to use this methodology to test if PumHD would bind preferentially to Xbp1^{spliced} or Xbp1^{unspliced}, which was not done in Gerber et al. The result is that PumHD binds equally well both Xbp1 forms.

2. *Figure 2c, are the PumHD and FullPum constructs used in this assay RNAi resistant?*

For the PumHD and FullPum overexpression constructs we used plasmids that lack the Pum 3'UTR region that is the targeted by Pum RNAi.

3. *What are the subcellular localization of Pum and Ire1 kinase, in the cytosol or in ER?*

It is known that both Ire1 kinase domain and Pumilio localize mostly in the cytosol.

4. *The authors claimed that "Indeed, Le Thomas et al showed that in its purified, fully phosphorylated oligomeric form hIRE1 α KR (KR-3P) degraded not only RIDD and RIDDLE mRNA targets, but also Xbp1 mRNA." And the made hypothesis on it. However, there is no data or reference supporting this notion.*

We now give the full reference for Le Thomas et al:

LeThomas, A., Ferri E., Wu T., Marsters, S., Harnoss JM, Modrusan Z., Chan S., Solon M., Chalouni C., Li, W., Koeppen H., Rudolph J., Wang W., Walter P., and Avi Ashkenazi A. (2021) Noncanonical mRNA decay by the endoplasmic-reticulum stress sensor IRE1 α promotes cancer cell survival.

This manuscript was co-submitted with ours to Nature Communications and is now accepted for publication in this journal. Since our initial submission this manuscript was also published in bioRxiv

<https://www.biorxiv.org/content/10.1101/2021.03.16.435520v1>

This claim regarding Le Thomas et al is for human XBP1, so we have corrected our sentence to reflect that: "Indeed, Le Thomas et al showed that in its purified, fully- phosphorylated oligomeric form hIRE1 α KR (KR- 3P) degraded not only RIDD and RIDDLE mRNA targets, but also hXBP1 mRNA".

Finally, we now present results in Supplementary Fig. 4a, showing that human PUM1 can protect human XBP1 from non-canonical IRE1 dependent mRNA decay. We also show in Sup Fig 4b that siRNAs for PUM1 and PUM2 in human MDAMB231 cells lead to a reduction in the levels of hXBP1 and the XBP1 target gene SYVN1, upon treatment with Thapsigargin, to induce ER stress. These results are similar to the results we obtained for *Drosophila* Xbp1 (Fig. 5).

Sincerely,

Pedro M. Domingos

Reviewers' Comments:

Reviewer #1:

Remarks to the Author:

The authors have addressed all my original comments.

I'd like to congratulate the authors on the excellent work.

Reviewer #2:

Remarks to the Author:

The authors have done a nice job at addressing my comments.

Reviewer #3:

Remarks to the Author:

I am satisfied with the revised version. The authors have addressed my major concerns.